# ColA: Collaborative Adaptation with Gradient Learning

## Abstract

A primary function of back-propagation is to compute both the gradient of hidden representations and parameters for optimization with gradient descent. Training large models requires high computational costs due to their vast parameter sizes. While Parameter-Efficient Fine-Tuning (PEFT) methods aim to train smaller auxiliary models to save computational space, they still present computational overheads, especially in Fine-Tuning as a Service (FTaaS) for numerous users. We introduce Collaborative Adaptation (ColA) with Gradient Learning (GL), a parameter-free, model-agnostic fine-tuning approach that decouples the computation of the gradient of hidden representations and parameters. In comparison to PEFT methods, ColA facilitates more cost-effective FTaaS by offloading the computation of the gradient to low-cost devices. We also provide a theoretical analysis of ColA and experimentally demonstrate that ColA can perform on par or better than existing PEFT methods on various benchmarks.

## 1 Introduction

Transfer learning with pretrained foundation models plays a crucial role in deep learning applications such as natural language processing and computer vision. Adapting these models to specific tasks via fine-tuning has become a prevalent paradigm (Peters et al., 2018; Fedus et al., 2022; Devlin et al., 2018). However, full Fine-Tuning (FT), which modifies all model parameters, becomes computationally prohibitive because each data task requires a unique set of fine-tuned parameters. The size of recent deep models has dramatically increased, ranging from hundreds of millions (Radford et al., 2019; Lewis et al., 2019) to hundreds of billions (Brown et al., 2020; Muennighoff et al., 2022) or even trillions (Fedus et al., 2022) of parameters. As these models continue to expand, developing efficient methods for adaptation and deployment becomes imperative. In response to these challenges, Parameter-Efficient Fine-Tuning (PEFT) has been introduced (Houlsby et al., 2019; Zaken et al., 2021; Li & Liang, 2021; Hu et al., 2021; He et al., 2021). Specifically, PEFT methods update only a small number of free parameters, with the amount less than 1% of the original model parameters, while keeping the pretrained parameters frozen. These methods can achieve comparable performance to full FT on various tasks and significantly reduce the computational cost (Hu et al., 2021; He et al., 2021).

In an era where personalized models are increasingly in demand, we aim to develop a system to provide Fine-Tuning as a Service (FTaaS) for numerous users. However, existing PEFT methods introduce significant computational overhead because each user would require a separate set of trainable parameters and their corresponding gradient for fine-tuning with gradient descent (Dettmers et al., 2023). Due to the constraints of computational space in edge devices, handling these extra parameters becomes challenging, especially when offering FTaaS to a large number of users. In this work, we introduce Collaborative Adaptation (ColA) with Gradient Learning (GL), a parameter-free and model-agnostic fine-tuning method that decouples the computation of the gradient of hidden representations and parameters. Our proposed method offers a more cost-efficient FTaaS system compared with PEFT methods, enhancing collaboration between the central server and local users. This efficiency improvement is achieved by offloading the computation of the parameter gradient to lower-cost devices, thereby conserving computational resources.

Our contributions are threefold:

- We introduce Gradient Learning (GL), a new learning framework based on functional gradient descent, specifically designed to decouple the computation of the gradient of hidden representations and parameters. A theoretical analysis of GL is provided, showcasing its equivalence to the classical gradient descent method.
- We conceptualize Fine-Tuning as a Service (FTaaS) and propose Collaborative Adaptation (ColA) with GL as a cost-effective solution for FTaaS by offloading the computation of the gradient to low-cost devices. We utilize the parameter merging technique to further reduce the cost of computation space and to integrate multiple adapters for collaboratively leveraging local computation resources.
- Through comprehensive experiments on diverse benchmarks, we demonstrate that ColA consistently matches or outperforms existing PEFT methods in performance. We conduct a comprehensive quantitative evaluation of computational costs on real devices. We also provide a theoretical analysis of parameter merging, present empirical results underscoring the benefits of user collaboration, and conduct ablation studies.

## 2 RELATED WORKS

**Parameter-Efficient Fine-Tuning (PEFT)** PEFT methods aim to fine-tune large pretrained models to downstream tasks. To achieve computational efficiency, they freeze the parameters of the pretrained network while fine-tuning a small set of tunable parameters. Previous works propose to insert Adapter layers between existing layers in the pretrained network (Rebuffi et al., 2017; Houlsby et al., 2019). However, this sequential nature can lead to inference latency, particularly with small batch sizes, as the Adapter layers have to be processed sequentially and the extra computation cannot be bypassed directly (Hu et al., 2021). In contrast, Prefix Tuning (Li & Liang, 2021) and Low-Rank Adaptation (LoRA) (Hu et al., 2021) offer parallel computations for inference. Prefix Tuning prepends a sequence of continuous vectors, referred to as the "prefix", to the input or hidden layers. This prefix can be seen as the tunable instruction prompting applied to the word embeddings. Nevertheless, it has been noted that Prefix Tuning suffers from optimization difficulty and lack of stability (Li & Liang, 2021; Hu et al., 2021). LoRA (Hu et al., 2021) introduces two trainable low-rank matrices to reparameterize the updates of pretrained weight matrices in downstream tasks. This approach avoids additional inference computation by merging the fine-tuned matrices together with the frozen pretrained weights. A unified framework of PEFT including Adapter (Houlsby et al., 2019), Prefix Tuning (Li & Liang, 2021), and LoRA (Hu et al., 2021) has been proposed by He et al. (2021). Specifically, it shows an alternative form of Prefix Tuning, revealing its close relationship with Adapter tuning, and then it conceptualizes PEFT as the process of learning a modification of hidden representations. Recent advances such as few-shot (Liu et al., 2022) and quantized (Dettmers et al., 2023) fine-tuning methods are proposed to further enhance the computational efficiency. Existing PEFT methods primarily focus on *modeling* various additional tunable structures to reduce computational cost on fine-tuning. In contrast, we propose a computationally efficient *learning* algorithm for fine-tuning large models for downstream tasks, which is, to our best knowledge, the first work on this focus.

**Functional gradient descent** Functional gradient descent generalizes gradient descent by optimizing a loss function within a function space rather than a parameter space. Mason et al. (1999) introduces an alternative view of boosting algorithms as iterative functional gradient descent algorithms, which has led to the recent advanced development of Gradient Boosting (GB) (Friedman, 2001) in the area of deep learning (Nitanda & Suzuki, 2018; Huang et al., 2018; Diao et al., 2022). TrAdaBoost (Dai et al., 2007) extends AdaBoost (Freund & Schapire, 1995) to transfer learning with classical machine learning models. It utilizes the boosting algorithm to learn a new classifier to adapt changes in the training data. Our work is primarily inspired by Gradient Assisted Learning (GAL) (Diao et al., 2022), which uses GB within a decentralized collaborative learning framework. It promotes collaboration in supervised learning without sharing local data, models, or objective functions. Existing functional gradient descent methods like GB and GAL train a *distinct* model for each *non-stochastic* boosting iteration, utilizing the "pseudo-residual" that approximates the gradient of the model's *output*. In contrast, our proposed method fine-tunes the *same* model at each *stochastic* boosting iteration using the gradient of *hidden representations*, so that an ensemble of weak learners is no longer needed. Furthermore, we concurrently apply stochastic functional gradient descent in multiple intermediate layers of pretrained large models and extend our algorithm to a distributed setting to promote collaboration among local users.

## 3 METHOD

### 3.1 PROBLEM

**Fine-Tuning (FT)** Consider a dataset of $N$ samples, denoted by $\mathcal{D} = \{(x_i, y_i)\}_{i=1}^{N}$, where $x_i$ is the input and $y_i$ is the corresponding target. Suppose the architecture of a deep neural network $f_\theta(\cdot)$ contains $M$ distinct layers, each represented by $f_{\theta_m}(\cdot)$ for $m = 1, \ldots, M$. For each layer $m$, the network processes a specific hidden input $x_{i,m}$ and consequently produces a hidden representation, denoted by $h_{i,m} = f_{\theta_m}(x_{i,m})$. In subsequent notations, we omit the index $i$ for clarity. Note that the hidden input $x_m$ for any layer $m$ refers to the hidden representation of its preceding layer. In the special case of the first layer, $x_{i,1}$ corresponds to the input data $x_i$ of dataset $\mathcal{D}$. Given a pretrained network $f_{\theta_{1:M}}(\cdot)$, the objective is to minimize its empirical risk in order to fine-tune the model, which can be represented by

$$\min_{\theta_{1:M}} \mathbb{E}_N \mathcal{L}(y, f_{\theta_{1:M}}(x)), \tag{1}$$

where $\mathcal{L}(\cdot)$ is the loss function and $\mathbb{E}_N$ denotes the empirical average over the dataset $\mathcal{D}$. A conventional fine-tuning method utilizes gradient descent for parameter optimization by computing the derivative of the loss with respect to the model parameters $\theta_{1:M}$, written as

$$\nabla \theta_{1:M} \triangleq \nabla_{\theta_{1:M}} \mathbb{E}_N \mathcal{L}(y, f_{\theta_{1:M}}(x)). \tag{2}$$

**Parameter-Efficient Fine-Tuning (PEFT)** Parameter-Efficient Fine-Tuning (PEFT) has been introduced to reduce the computational cost of fine-tuning large models such as Large Language Models (LLM). PEFT methods typically incorporate an auxiliary model, denoted as $g_{w_m}(\cdot)$, which is parameterized by $w_m$ for each layer $m$. During the fine-tuning process, PEFT methods freeze the original model parameters $\theta_{1:M}$ and only update the auxiliary parameters $w_{1:M}$. A notable advantage of this approach is that the dimensionality of $w_{1:M}$ tends to be considerably smaller than that of $\theta_{1:M}$, leading to substantial savings in storage requirements of back-propagation, especially in the memory usage of Graphics Processing Units (GPU). Low-Rank Adaptation (LoRA), one of the most widely used PEFT methods, ingests the hidden input $x_{i,m}$ and produces a change of hidden representation $\Delta h_{i,m}$. The fine-tuned hidden representation $\hat{h}_{i,m}$ will be used in the original network as a fine-tuned replacement of $h_m$ as described by

$$\Delta h_{i,m} = g_{w_m}(x_{i,m}), \quad \hat{h}_{i,m} = h_{i,m} + \alpha \cdot \Delta h_{i,m},$$

where $\alpha$ is a scaling factor that can be tuned during inference. The fine-tuned model is denoted by $f_\theta(x, \Delta h_{1:M})$. Given a pretrained network $f_{\theta_m}(\cdot)$, the objective of LoRA becomes $\min_{w_{1:M}} \mathbb{E}_N \mathcal{L}(y, f_\theta(x, \Delta h_{1:M}))$, and it can be optimized by computing the derivative of the loss with respect to the auxiliary parameters $w_{1:M}$ as follows

$$\nabla w_{1:M} \triangleq \nabla_{w_{1:M}} \mathbb{E}_N \mathcal{L}(y, f_\theta(x, \Delta h_{1:M})). \tag{3}$$

### 3.2 COLLABORATIVE ADAPTATION

We propose Collaborative Adaption (ColA) as a framework for providing Fine-Tuning as a Service(FTaaS) with our novel Gradient Learning (GL) algorithm. GL is both *parameter-free* and *model-agnostic* because it decouples the computation of the gradient of auxiliary parameters from that of the fine-tuned hidden representations, a process we refer to as *Gradient Decoupling*. Meanwhile, our method can significantly improve computational efficiency and provide cost-effective service for many users by offloading the computation of the gradient to low-cost devices.

**Gradient Learning (GL)** We propose Gradient Learning (GL) in order to save the storage requirements of back-propagation. Existing PEFT methods focus on optimizing the parameter efficiency of the auxiliary model $g_{w_m}(\cdot)$ to achieve the same goal. Specifically, PEFT methods fine-tune the pretrained model for downstream tasks by leveraging a set of auxiliary parameters, denoted as $w_{1:M}$, while keeping the original parameters $\theta_{1:M}$ frozen. Unlike full FT, this strategy bypasses the computation of $\nabla \theta_{1:M}$ during the back-propagation phase. Efficiency is further enhanced by minimizing the size of these auxiliary parameters. Without explicitly optimizing $\theta_{1:M}$, it has been shown that optimizing $w_{1:M}$ can still achieve satisfactory results (Hu et al., 2021; He et al., 2021). In order

to optimize the auxiliary parameters $w_{1:M}$, the back-propagation procedure of PEFT methods computes $\nabla w_{1:M}$, and consequently, the gradient of fine-tuned hidden representations through chain rule, denoted as

$$\nabla \hat{h}_{1:M} \triangleq \nabla_{\hat{h}_{1:M}} \mathbb{E}_N \mathcal{L}(y, f_\theta(x, \Delta h_{1:M})). \qquad (4)$$

On the contrary, GL is a new learning algorithm that decouples the computation of the gradient of auxiliary parameters $\nabla w_{1:M}$ and fine-tuned hidden representations $\nabla \hat{h}_{1:M}$. At the beginning, we forward data $x$ and the change of hidden representations $\Delta h_{1:M}$ into both the pretrained base model and the newly initialized auxiliary models to obtain the model output $f_\theta(x, \Delta h_{1:M})$ and fine-tuned hidden representations $\hat{h}_{1:M}$. Then, we compute the gradient of fine-tuned hidden representations $\nabla \hat{h}_{1:M}$ which naturally exist in the back-propagation stage of deep neural networks. Meanwhile, if $\alpha = 1$ and using $\frac{\partial \hat{h}_{1:M}}{\partial \Delta h_{1:M}} = \alpha$, we have the gradient of fine-tuned hidden representations equal to the gradient of the change of hidden representation as follows

$$\nabla \hat{h}_{1:M} = \nabla_{\Delta h_{1:M}} \mathbb{E}_N \mathcal{L}(y, f_\theta(x, \Delta h_{1:M})). \qquad (5)$$

It is essential to note that during the back-propagation stage, GL does not compute the gradient of either the original or the auxiliary parameters. After completing the forward and backward propagation stages, we can transfer the hidden inputs $x_{1:M}$ and the gradient of the fine-tuned hidden representations $\nabla \hat{h}_{1:M}$ to multiple low-cost devices, such as Central Processing Unit (CPU) or low-end GPUs. By doing so, we reduce the cost of computational space on resource-intensive devices which host the large base model. We refer to this process as *Gradient Offloading*. Our proposed approach offers a significant computational advantage, because the computational space of the GPU hosting the large base model is considerably more valuable than that of low-end GPUs, CPUs and storage drives. Furthermore, computing the gradient of auxiliary parameters on low-cost devices will not disrupt the computation of the gradient of hidden representations on the GPU, where the large base model is hosted. As a result, we can run two decoupled gradient computations in parallel for different batches of data.

Next, we can optimize the auxiliary models $x_{1:M} \mapsto g_{w_{1:M}}(x_{1:M})$ in parallel on multiple low-cost device. We achieve this by fitting the target $\Delta h_m - \nabla \hat{h}_m$, which are calculated from the last update and treated as a fixed term. For instance, we define the auxiliary quadratic loss as follows

$$\ell_m(x, y; w_m) \triangleq \frac{1}{2} \|g_{w_m}(x_m) - (\Delta h_m^t - \nabla \hat{h}_m^t)\|_2^2, \text{ where} \qquad (6)$$

$$\Delta h_m^t \triangleq g_{w_m^t}(x_m), \quad \nabla \hat{h}_m^t \triangleq \left. \frac{\partial \mathcal{L}(y, f_\theta(x, \Delta h_{1:M}^t))}{\partial \hat{h}_m} \right|_{\hat{h}_m = h_m + \Delta h_m^t}$$

and $w_m^t$ represents the most recent estimate of $w_m$ at round $t$. We then operate gradient-based optimizations using (6). Its validity is justified by the following result.

**Proposition 1** *The gradient $\nabla_{w_m} \ell_m(x, y; w_m)$ and $\nabla_{w_m} \mathcal{L}(y, f_\theta(x, \Delta h_{1:M}))$ evaluated at $w_m = w_m^t$ are the same for any $w_m^t$.*

The gradient of the updated auxiliary model $g_{w_m}(\cdot)$ essentially moves $w_m$ toward the optimal direction of $h_m$ that minimizes the objective loss. Intuitively, we reconstruct the computation graph of the auxiliary model $g_{w_m}(\cdot)$ with the hidden input $x_m$ and optimize the auxiliary parameters $w_m$ with the change of hidden representations $\Delta h_m$ and the gradient of fine-tuned representations $\nabla \hat{h}_m$. It is essential to note that the optimization of $g_{w_m}(\cdot)$ is decoupled from the computation of $\nabla \hat{h}_{1:M}$. In practice, we can optimize $g_{w_{1:M}}(\cdot)$ by taking one step of gradient descent with a learning rate $\gamma$ on multiple low-cost devices in parallel without interfering with the computation of $\nabla \hat{h}_{1:M}$ on the high-cost GPU hosting the large base model.

Existing learning algorithms compute the gradient of hidden representations and parameters concurrently during back-propagation. However, this classical approach is significantly limited by the computation space of the high-cost device, which further restricts the training batch size, thus affecting the overall training time of deep neural networks. Our approach demonstrates that we can

train the network by computing the gradient of hidden representations and parameters separately. Additionally, storage limitations also affect the choice of auxiliary models, impacting fine-tuning performance. Our method allows for a broader selection of auxiliary models, as their optimization can be done on a different device. Theoretically, our proposed Gradient Learning (GL) is related to functional gradient descent and Gradient Boosting (GB) (Friedman, 2001). Our work is primarily inspired by Gradient Assisted Learning (GAL), a recent advance of functional gradient descent in distributed machine learning (Diao et al., 2022). While methods like GB and GAL use a unique model for every non-stochastic boosting iteration at the output layer, our GL method retrains the same model for every stochastic boosting iteration at intermediate layers.

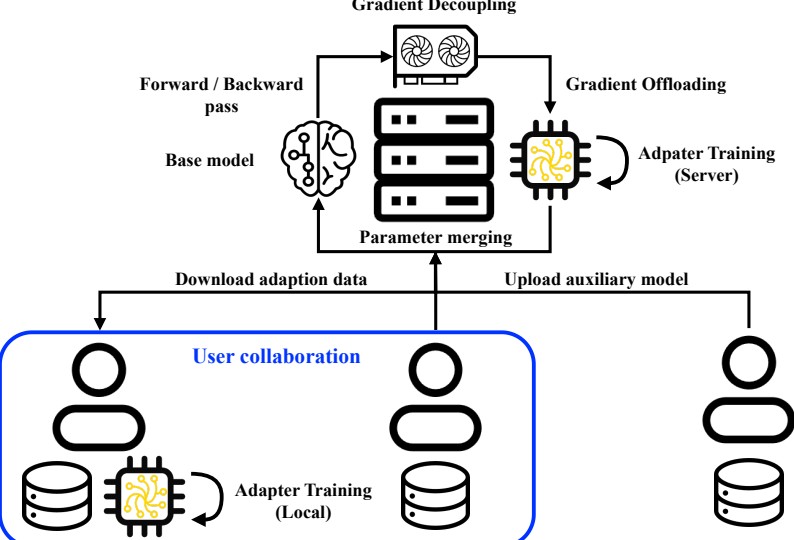

Figure 1: Illustration of the Fine-Tuning as a Service (FTaaS) system architecture. A central server handles both forward and backward passes of the pretrained model. It offloads gradient computations to a low-cost device, namely, *Gradient Offloading*. Meanwhile, adapters can be trained either on the server or locally by downloading adaptation data. Users of FTaaS can train their adapters independently or collaborate with others if needed.

**Fine-Tuning as a Service (FTaaS)**   As illustrated in Figure 1, we propose a computationally scalable approach for Fine-Tuning as a Service (FTaaS) designed to prevent the cost of computational space on the GPU from increasing proportionally with the number of users of the service. Notably, given the constraints of computational space when fine-tuning large models on GPU, we introduce Collaborative Adaptation (ColA) with GL through a stochastic optimization procedure as detailed in Algorithm 1. During each training iteration $t$, a batch of data from $K$ users $(x_{1:K}^t, y_{1:K}^t)$ of size $B$ is sampled from the training dataset $(x_{1:K}, y_{1:K})$. At $t = 1$, the auxiliary parameters $w_{1:M,1:K}$ are initialized to zero. After receiving $M \times K$ auxiliary models from $K$ collaborative users, we can compute the output of the model $f_\theta(x_{1:M,1:K}^t, \Delta h_{1:M,1:K}^t)$, where $\Delta h_{m,k}^t \triangleq g_{m,k}^t(x_{m,k}^t)$. Then, we gather the hidden input of auxiliary models $x_{1:M}^t$ during the forward pass, achievable via the `hook` functionality in Pytorch (Paszke et al., 2019). Next, we compute the backward pass of the loss $\mathcal{L}(y_{1:K}^t, f_\theta(x_{1:M,1:K}^t, \Delta h_{1:M,1:K}^t))$ on GPU, transfer and save the hidden input of auxiliary models $x_{1:M,1:K}^t$ and the gradient of fine-tuned hidden representations $\nabla \hat{h}_{1:M,1:K}^t$ to buffers on low-cost devices. Due to the computational space constraints of batch size $B$ on the GPU, we use buffers to accumulate adequate adaptation data over $I$ batches, where $I$ is referred to as the adaptation interval in Algorithm 1. By increasing $I$, we can effectively increase the batch size, which may allow for faster model convergence due to a more accurate estimate of the gradient of the auxiliary parameters. Once the buffers collect sufficient adaptation data, we can optimize each auxiliary model $g_{w_{m,k}}^t$ in parallel following Equation 6. Similar to ZeRO-Offload (Ren et al., 2021), our method can also save the state information of adaptive optimizers like Adam (Kingma & Ba, 2014) on low-cost devices. Notably, each training iteration only involves one forward and backward pass on GPU, which is the

same as the classical gradient descent procedure. The distinction is that we decouple and offload the computation of the gradient of parameters to separate devices.

---

**Algorithm 1** ColA: Collaborative Adaptation with Gradient Learning

---

**Require:** Training data set $x$ and target $y$, a base model $f_\theta(\cdot)$, $M$ fine-tuning layers, $K$ collaborative users, $M \times K$ auxiliary models $g_{w_{1:M,1:K}}(\cdot)$, the number of training iterations $T$, the loss function $\mathcal{L}$, the training batch size $B$, the learning rate $\gamma$, and the adaptation interval $I$.

1: **for** each training iteration $t$ from 1 to $T$ **do**
2:     $(x_{1:K}^t, y_{1:K}^t) \leftarrow$ Sample a batch of size $B$ from the training dataset $(x_{1:K}, y_{1:K})$
3:     (Optional) Merge $M \times K$ auxiliary models of $K$ users to the base model
4:     Compute forward pass of $f_\theta(x_{1:M,1:K}^t, \Delta h_{1:M,1:K}^t)$, with $\Delta h_{m,k}^t \leftarrow g_{m,k}^t(x_{m,k}^t)$
5:     Gather hidden input of auxiliary models $x_{1:M,1:K}^t$ from forward pass
6:     Compute backward pass of $\mathcal{L}(y_{1:K}^t, f_\theta(x_{1:M,1:K}^t, \Delta h_{1:M,1:K}^t))$
7:     Gather gradient of hidden representations
        $\nabla \hat{h}_{1:M,1:K}^t = \nabla_{\hat{h}_{1:M,1:K}^t} \mathcal{L}(y_{1:K}^t, f_\theta(x_{1:M,1:K}^t, \Delta h_{1:M,1:K}^t))$
8:     (Optional) Unmerge $M \times K$ auxiliary models of $K$ users from the base model
9:     Transfer $(x_{1:M,1:K}^t, \nabla \hat{h}_{1:M,1:K}^t)$ to low-cost devices
10:    **for** each adapter $m$ of user $k$ *in parallel* **do**
11:        Save adaptation data $(x_{m,k}^t, \nabla \hat{h}_{m,k}^t)$ to buffer
12:        **if** $t \bmod I = 0$ **then**
13:            Compute forward pass $\Delta h_{m,k}^{t-I:t} = g_{w_{m,k}}^t(x_{m,k}^{t-I:t})$
14:            Optimize $g_{w_{m,k}}^t(\cdot)$ with $(x_{m,k}^{t-I:t}, \Delta h_{m,k}^{t-I:t} - \nabla \hat{h}_{m,k}^{t-I:t})$ and learning rate $\gamma^t$
15:            Transfer auxiliary model $g_{w_{n,k}}^t(\cdot)$ to the server
16:            Empty buffer

---

**Parameter merging**   We study and integrate parameter merging into our algorithm to further reduce the cost of computation space. Previous work has demonstrated that the original parameters $\theta_m$ and the auxiliary parameters $w_m$ can be merged into one network after fine-tuning is finished for inference (He et al., 2021; Mangrulkar et al., 2022). PEFT methods like LoRA offer such capability, and its tuning factor $\alpha$ can adjust the contribution of the adapter to the output of the model. Additionally, this approach can also combine multiple adapters trained on different datasets. We propose to optionally utilize parameter merging during training, as shown in Algorithm 1. It allows $K$ users to collaboratively fine-tune the auxiliary models with their local data and computation resources. A key advantage of this method is eliminating the need to iterate multiple adapters and save the computation space of auxiliary parameters and hidden representations in the forward pass and their corresponding gradient in the backward pass, as depicted in Table 1. In fact, by utilizing parameter merging, our method consumes the same amount of memory for a given batch size, regardless of the size of auxiliary models and the number of users, because the computation related to auxiliary models has been completely offloaded to low-cost devices. To the best of our knowledge, despite being a widely used technique, no previous work has explicitly studied the requirements of parameter merging. For any layer $m$ to be fine-tuned, one wishes that the fine-tuned auxiliary model $g_{w_m}(\cdot)$ can be merged back to the original model architecture to simplify computation. The following result shows that $g_{w_m}(\cdot)$ must be linear in the input.

**Proposition 2** *Consider a linear function $x \mapsto f_\theta(x) = \theta x$, where $\theta \subset \Theta \subseteq \mathbb{R}^{d_1 \times d_2}$ is the parameter and $x \in \mathbb{R}^{d_2}$. Assume that $g: \mathbb{R}^{d_2} \mapsto \mathbb{R}^{d_1}$ is such a function that $x \mapsto f_\theta(x) + g(x)$ can be equivalently written as $x \mapsto f_{\hat{\theta}}(x)$ for some $\hat{\theta} \in \Theta$. Then, $g$ must be a linear function of $x$ and written as $wx$ for some $w \in \mathbb{R}^{d_2 \times d_1}$.*

**Remark 1** *In the result above, $f_\theta(\cdot)$ represents the function of a generic layer, and $g(\cdot)$ denotes the auxiliary model. The parameterization of $g(\cdot)$ is not necessarily through $w$, as it could be a smaller parameter that maps to $w$. An example is the low-rank approximation $w = w_1 \cdot w_2$ where $w_1$ has a small number of hidden sizes.*

In Table 1, we compare the complexity of the computation space of FT, PEFT, and ColA. Existing PEFT methods require significantly less computational space compared to FT, primarily because the

size of auxiliary parameters $w_{1:m,1:K}$ and their gradient $\nabla w_{1:M,1:K}$ is minimal. Nevertheless, PEFT methods still require the computation of $\nabla w_{1:M,1:K}$ together with $\nabla h_{1:M}, \nabla \tilde{h}_{1:M,1:K}$, because they utilize the classical gradient descent learning procedure. In contrast, our proposed method is *parameter-free* because it completely decouples the computation related to auxiliary parameters $\nabla w_{1:M,1:K}$ from the forward and backward pass of the base model. If the parameter merging technique was used, ColA (merged) only computes the gradient of hidden representations $\nabla h_{1:M}$ on GPU. As a result, ColA (merged) can even reduce the cost of full fine-tuning by offloading all other computations to low-cost devices. To our knowledge, this has not been achieved with any existing methods. It indicates that our method can achieve the performance of full parameter training from scratch while reducing the computation space bottleneck. As depicted in Table 1, PEFT methods would multiply the cost of computational space on the GPU by $K$, whereas ColA (merged) incurs no additional overhead on the GPU. Furthermore, our method can effectively leverage distributed low-cost devices to optimize the auxiliary models in parallel. Thus, our method emerges as a more cost-effective and scalable option for FTaaS compared with existing PEFT methods. While our approach introduces additional runtime due to the transmission of adaptation data and auxiliary models from one device to another, we consider this a technical limitation, which could potentially be mitigated with further engineering refinements. For example, we can instead offload the computation to low-end GPUs instead of CPUs to speed up the transmission.

Table 1: The complexity of computation space of FT, PEFT, and ColA. Representations, parameters and their gradient in $\{\cdot\}$ can be stored in low-cost devices. $h_{1:M,1:K}$ denotes hidden representations of the base model and $\tilde{h}_{1:M,1:K}$ denotes the hidden representations of the auxiliary models.

| Method | | | Forward | | Backward | |
|---|---|---|---|---|---|---|
| | | | Representations | Parameters | Representations | Parameters |
| FT | | Inference | — | $\theta_{1:M}$ | — | |
| | | Learning | $h_{1:M}$ | $\theta_{1:M}$ | $\nabla h_{1:M}$ | $\nabla\theta_{1:M}$ |
| PEFT | unmerged | Inference | — | $\theta_{1:M}, w_{1:M,1:K}$ | — | |
| | | Learning | $h_{1:M}, \tilde{h}_{1:M,1:K}$ | $\theta_{1:M}, w_{1:M,1:K}$ | $\nabla h_{1:M}, \nabla\tilde{h}_{1:M,1:K}$ | $\nabla w_{1:M,1:K}$ |
| | merged | Inference | — | $\theta_{1:M}$ | — | |
| ColA | unmerged | Inference | — | $\theta_{1:M}, w_{1:M,1:K}$ | — | |
| | | Learning | $h_{1:M}, \tilde{h}_{1:M,1:K}$ | $\theta_{1:M}, w_{1:M,1:K}$ | $\nabla h_{1:M}, \nabla\tilde{h}_{1:M,1:K}$ | $\{\nabla w_{1:M,1:K}\}$ |
| | merged | Inference | — | $\theta_{1:M}$ | — | |
| | | Learning | $h_{1:M}, \{\tilde{h}_{1:M,1:K}\}$ | $\theta_{1:M}, \{w_{1:M,1:K}\}$ | $\nabla h_{1:M}, \{\nabla\tilde{h}_{1:M,1:K}\}$ | $\{\nabla w_{1:M,1:K}\}$ |

Our method, rooted in a functional gradient descent learning framework, is notably *model-agnostic*. The choice of auxiliary models $g_{w_{1:M,1:K}}(\cdot)$ is independent of the base model $f_\theta(\cdot)$. Moreover, different model architectures can be utilized for auxiliary models at various layers. It is crucial to recognize that the architecture of these auxiliary models significantly influences performance. For example, rather than using low-rank approximations like LoRA, it is feasible to utilize a linear layer or even a Multi-Layer Perceptron (MLP), because the computation of these auxiliary models is decoupled from that of the base model and offloaded to other devices. Also, interactive auxiliary models, such as those in ControlNet (Zhang & Agrawala, 2023), where one model's output serves as another's input, may also utilize the proposed method. As depicted in Figure 1, another advantage of our model-agnostic approach is that the users of FTaaS can locally fine-tune their adapters using adaptation data received from the server if they have computational resources. Additionally, based on their available computational resources, users have the flexibility to customize the optimization of their adapters.

# 4 EXPERIMENTAL STUDIES

## 4.1 EXPERIMENTAL SETUP

We compare ColA with full fine-tuning (FT) and PEFT methods including LoRA (Hu et al., 2021), AdaLoRA (Zhang et al., 2023), IA3 (Liu et al., 2022), Prompt Tuning (Lester et al., 2021), Prefix Tuning (Li & Liang, 2021), and P-Tuning (Liu et al., 2023). We conduct experiments on three tasks

including Sequence Classification (SC), Sequence to Sequence (S2S), and Causal Language Modeling (CLM). For the SC task, we use RoBERTa (base) (Liu et al., 2019) as the pretrained model with GLUE benchmark for all methods (Wang et al., 2018). In the S2S task, we use BART (base) (Lewis et al., 2019) as the pretrained model and evaluate all methods with a range of datasets including Financial Phrase Bank (FPB) (Malo et al., 2014), WikiSQL (Zhong et al., 2017), SAMSum (Gliwa et al., 2019), E2E NLG (Dušek et al., 2020), WebNLG (Gardent et al., 2017), and DART (Nan et al., 2020). For the CLM task, we utilize GPT-2 (Radford et al., 2019) and Llama-2 (Touvron et al., 2023) as the pretrained model with the Dolly (Conover et al., 2023) dataset to perform instruction tuning (Wei et al., 2021).

We conduct experiments using Low Rank, Linear, and Multi-Layer Perceptron (MLP) auxiliary models to demonstrate the model-agnostic nature of our proposed ColA method. For the low-rank variant of ColA, termed as ColA (Low Rank), we use a hidden dimension $r = 8$, identical to those of LoRA and AdaLoRA to ensure a consistent evaluation across methods. Our Linear auxiliary model has parameters that match the count of the weight matrix it fine-tunes, while our MLP configuration uses a two-layer neural network with a hidden size of 128 and ReLU activation. We apply the default configurations from the PEFT package (Mangrulkar et al., 2022) for other PEFT baselines. We demonstrate the hyperparameters used in our experiments in Table 5. Further experimental results including learning from scratch, adaptation interval, computation evaluation, and learning curves are available in the Appendix.

## 4.2 EXPERIMENTAL RESULTS

We demonstrate the results of RoBERTa (base) on the SC task with GLUE metric in Table 2 and the results of BART (base) on the S2S task with ROUGE (Longest) metric in Table 3. Compared with the existing PEFT methods, ColA consistently outperforms PEFT methods including IA3, Prompt Tuning, Prefix Tuning, and P-Tuning. Furthermore, ColA outperforms AdaLoRA on average for both SC and S2S tasks with fewer trainable parameters. Meanwhile, ColA performs on par with LoRA across most datasets.

Table 2: Results of RoBERTa (base) on the Sequence Classification task with GLUE metric. The gradient of parameters in $\{\cdot\}$ can be stored in low-cost devices. Both parameters and their gradient in $[\cdot]$ can be stored in low-cost devices.

| Method | | Trainable Parameters | MNLI | SST-2 | MRPC | CoLA | QNLI | QQP | RTE | STS-B | Avg. |
|---|---|---|---|---|---|---|---|---|---|---|---|
| FT | | 125.2 M (100.0 %) | 87.2 | 95.0 | 89.3 | 61.8 | 93.0 | **91.8** | **76.0** | 90.4 | 85.6 |
| LoRA | | 887.0 K (0.7 %) | 86.7 | 95.1 | 89.4 | 62.8 | 93.0 | 90.9 | 75.1 | 90.3 | 85.4 |
| AdaLoRA | | 11.3 M (9.0 %) | 87.5 | 95.2 | 88.2 | 60.3 | 93.1 | 91.4 | 65.3 | 90.2 | 83.9 |
| IA3 | | 629.0 K (0.5 %) | 85.1 | 93.8 | 83.2 | 52.6 | 91.4 | 88.7 | 65.7 | 87.5 | 81.0 |
| Prompt Tuning | | 607.5 K (0.5 %) | 79.2 | 93.0 | 75.7 | 16.2 | 86.8 | 82.9 | 57.9 | 62.6 | 69.3 |
| Prefix Tuning | | 960.8 K (0.8 %) | 85.3 | 93.2 | 72.3 | 23.4 | 90.6 | 88.9 | 61.7 | 66.9 | 72.8 |
| P-Tuning | | 821.5 K (0.7 M %) | 80.6 | 93.5 | 77.1 | 27.8 | 89.0 | 84.4 | 59.1 | 85.4 | 74.6 |
| ColA (Low Rank) | unmerged | {887.0 K (0.7 %)} | 86.9 | 94.6 | 90.4 | 61.1 | **93.2** | 90.9 | 73.6 | 90.2 | 85.1 |
| | merged | [887.0 K (0.7 %)] | 86.7 | 94.5 | 88.5 | 62.3 | 93.0 | 90.8 | 75.1 | 90.1 | 85.1 |
| ColA (Linear) | unmerged | {14.7 M (11.8 %)} | **87.2** | 95.4 | 88.5 | 61.3 | 92.9 | 90.9 | 71.8 | 90.9 | 84.9 |
| | merged | [14.7 M (11.8 %)] | 87.1 | **95.4** | 88.7 | 59.4 | 92.9 | 90.9 | 72.2 | **91.0** | 84.7 |
| ColA (MLP) | unmerged | {8.5 M (6.7 %)} | 86.9 | 95.1 | **90.4** | **64.6** | 92.4 | 91.3 | 73.6 | 90.7 | **85.6** |

It is worth noting that fine-tuning for SC requires training the classifier layers from scratch due to distinct target classes across datasets. While LoRA typically fine-tunes these classifier layers in conjunction with the auxiliary models, Gradient Learning (GL) computes the gradient of hidden representations during the backward pass of the base model. Consequently, we use a 'Linear' auxiliary model to train the newly initialized classifier layers. The results in Table 2 and 3 demonstrate that ColA (Low Rank) methods closely align with LoRA in performance, as the gradient computed with our methods exactly matches the gradient of LoRA. In our evaluations, we also compare different auxiliary models. The results demonstrate that the selection of an auxiliary model can influence the model performance, as ColA (Linear) and ColA (MLP) can outperform ColA (Low Rank). Notably, our proposed method can fine-tune without low-rank approximation while not incurring any additional cost of computation space, because the computation has been offloaded to separate low-cost devices.

Table 3: Results of BART (base) on the Sequence to Sequence task with ROUGE (Longest) metric. The gradient of parameters in $\{\cdot\}$ can be stored in low-cost devices. Both parameters and their gradient in $[\cdot]$ can be stored in low-cost devices.

| Method | | Trainable Parameters | FPB | WikiSQL | SAMSum | E2E NLG | WebNLG | DART | Avg. |
|---|---|---|---|---|---|---|---|---|---|
| FT | | 139.4 M (100.0 %) | 95.6 | 94.8 | 39.6 | 51.0 | 63.5 | 54.8 | 66.6 |
| LoRA | | 442.4 K (0.3 %) | 96.5 | 94.9 | 39.0 | 50.9 | 62.7 | 54.5 | 66.4 |
| AdaLoRA | | 13.0 M (9.3 %) | 93.8 | 95.4 | 38.7 | 50.9 | 63.4 | **55.1** | 66.2 |
| IA3 | | 36.9 K (0.03 %) | 71.4 | 86.0 | 34.6 | 49.7 | 53.9 | 49.9 | 57.6 |
| Prompt Tuning | | 30.7 K (0.02 %) | 71.8 | 75.8 | 32.1 | 44.0 | 40.2 | 38.9 | 50.5 |
| Prefix Tuning | | 184.3 K (0.1 %) | 75.8 | 83.6 | 27.7 | 33.6 | 33.9 | 33.1 | 47.9 |
| P-Tuning | | 244.7 K (0.2 %) | 82.8 | 77.7 | 31.6 | 40.8 | 33.3 | 37.3 | 50.6 |
| ColA (Low Rank) | unmerged | {442.4 K (0.3 %)} | 95.6 | 94.8 | 38.9 | 50.9 | 62.4 | 54.7 | 66.2 |
| | merged | [442.4 K (0.3 %)] | 96.5 | 94.7 | 38.7 | **51.0** | 62.5 | 54.6 | 66.3 |
| ColA (Linear) | unmerged | {21.2 M (15.2 %)} | 96.9 | 95.4 | 39.4 | 50.5 | 63.1 | 54.6 | 66.6 |
| | merged | [21.2 M (15.2 %)] | **98.2** | **95.6** | 39.0 | 50.8 | 63.1 | 55.0 | 66.9 |
| ColA (MLP) | unmerged | {11.8 M (8.5 %)} | 98.2 | 95.5 | **39.9** | 51.0 | **63.5** | 54.7 | **67.1** |

We demonstrate the results of user collaboration in Table 4. In the 'Joint' setup, all data is jointly trained with the same set of auxiliary models. The 'Alone' setup trains each data subset without merging the auxiliary models. In contrast, the 'Collaboration' setup merges auxiliary parameters together. Notably, users have the flexibility to determine local model architecture. For instance, ColA (Low Rank-Linear) indicates that the first four subsets (Classification, Information Extraction, Summarization, and Brainstorming) utilize ColA (Low Rank), while the remaining subsets use ColA (Linear). Results indicate that models trained in the 'Collaboration' setup perform on par with the 'Joint' and 'Alone' setup. However, the performance of 'Alone' diminishes after merging for inference, because 'Alone' setup does not utilize parameter merging during training.

Table 4: Results of GPT-2 on the Causal Language Modeling task with user collaboration. The gradient of parameters in $\{\cdot\}$ can be stored in low-cost devices. Both parameters and their gradient in $[\cdot]$ can be stored in low-cost devices.

| | Method | | Trainable Parameters | Classification | Information Extraction | Summarization | Brainstorming | Creative Writing | Open Q&A | Closed Q&A | General Q&A | All unmerged | All merged |
|---|---|---|---|---|---|---|---|---|---|---|---|---|---|
| Joint | Low Rank | unmerged | {294.9 K (0.2 %)} | 19.9 | 12.4 | 16.5 | 13.2 | 14.5 | 14.9 | 15.0 | 16.6 | 15.5 | 15.5 |
| | | merged | [294.9 K (0.2 %)] | 19.6 | 12.8 | 16.7 | 13.5 | **14.8** | 14.7 | 15.0 | **16.7** | 15.6 | |
| | Linear | unmerged | {21.2 M (17.1 %)} | 21.8 | 13.3 | **17.2** | 14.0 | 14.1 | 15.5 | 15.3 | 16.3 | 16.1 | 16.1 |
| | | merged | [21.2 M (17.1 %)] | **22.6** | **13.4** | 17.0 | **14.9** | 13.9 | **15.5** | **15.4** | 16.6 | **16.4** | |
| | MLP | unmerged | {8.7 M (7.0 %)} | 21.8 | 13.3 | 17.2 | 14.0 | 14.1 | 15.5 | 15.3 | 16.3 | 15.7 | — |
| Alone | Low Rank | | 8 × {294.9 K} | 19.1 | 13.5 | 15.3 | 13.6 | 14.6 | 15.2 | 16.1 | 16.2 | 15.7 | 13.9 |
| | Low Rank-Linear | | 4 × {294.9 K}, 4 × {21.2 M} | 18.9 | 11.2 | 15.7 | 13.7 | 13.7 | 15.5 | 16.3 | 16.2 | 15.5 | 13.9 |
| | Low Rank-MLP | | 4 × {294.9 K}, 4 × {8.7 M} | 21.4 | 11.8 | 15.5 | 13.4 | 14.0 | 15.3 | 16.9 | 16.4 | 15.9 | — |
| Collaboration | Low Rank | | 8 × [294.9 K] | 18.5 | 12.8 | 16.2 | 13.5 | 14.2 | 14.9 | 14.4 | 16.3 | 15.2 | |
| | Low Rank-Linear | | 4 × [294.9 K], 4 × [21.2 M] | 17.7 | 13.0 | 16.6 | 13.1 | 14.2 | 15.2 | 15.3 | 16.7 | 15.4 | |

We present ablation studies regarding the adaptation interval $I$ in Section C.4. For these experiments, we use a batch size of $B = 8$. By increasing the adaptation interval, such as $I = 4$, the effective batch size becomes $B \times I$. The results indicate that it is possible to achieve satisfactory convergence with fewer updates to the auxiliary models. This extension becomes especially valuable in situations demanding extensive computational space for computing the gradient of hidden representations for numerous users of FTaaS.

## 5 CONCLUSION

In this work, we address the pressing challenge of efficiently fine-tuning pretrained models for downstream tasks without incurring prohibitive computational costs. As pretrained models continue to grow in size and complexity, classical fine-tuning techniques have shown their limitations, especially when providing Fine-Tuning as a Service (FTaaS). We introduce Collaborative Adaptation (ColA) with Gradient Learning (GL), a parameter-free, model-agnostic fine-tuning approach that decouples the computation of the gradient of hidden representations and parameters and offloads the computation of the gradient to low-cost devices. We provide theoretical analysis and conduct extensive experiments to demonstrate that our method can perform on par or better than existing PEFT methods on various benchmarks with much less computation space bottleneck. Future works can further optimize the efficiency of the proposed method and broaden the application scope of FTaaS.

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

# Appendix

## A  LIMITATION AND FUTURE WORKS

While our method decouples and offloads gradient computation, it requires the transfer of auxiliary model from the user to the server and adaptation data from the server to the user. This transfer saves computational space on the high-cost device hosting the large base model but introduces an additional run time due to data transmission. We consider this a technical limitation that could be addressed with engineering refinements, especially at the system level design of FTaaS.

Enhancing the computational efficiency of our method is another avenue for future work. Integrating strategies from efficient and distributed machine learning, such as quantization (Dettmers et al., 2023), pruning (Diao et al., 2023), and Federated Learning (Diao et al., 2020), may be beneficial. Another exciting direction for GL could involve more advanced gradient decoupling on the hidden representations (Huang et al., 2018), paving the way for extensive model parallelization to improve run-time efficiency. Moreover, integrating interactive auxiliary models with ColA (Zhang & Agrawala, 2023) also represents a promising direction. Lastly, the applications of FTaaS could be further expanded by utilizing large language models (Touvron et al., 2023) and diffusion models (Rombach et al., 2022).

## B  THEORETICAL ANALYSIS

**Proposition 1**: The gradient $\nabla_{w_m} \ell_m(x, y; w_m)$ and $\nabla_{w_m} \mathcal{L}(y, f_\theta(x, \Delta h_{1:M}))$ evaluated at $w_m = w_m^t$ are the same for any $w_m^t$.

**Proof 1 (Proof of Proposition 1)** *Suppose we evaluate the gradient of $\ell_m(x, y; w_m)$ and $\mathcal{L}(y, f_\theta(x, \Delta h_{1:M}))$ at $w_m^t$, which can be an arbitrary value at round $t$. By the definition of $\ell_m$ and the chain rule, we have*

$$\frac{\partial \ell_m}{\partial w_m} = \frac{\partial \ell_m}{\partial g_{w_m}} \frac{\partial g_{w_m}}{\partial w_m} = (g_{w_m}(x_m) - (\Delta h_m - \nabla \hat{h}_m)) \frac{\partial g_{w_m}}{\partial w_m}. \tag{7}$$

*Evaluating the above equality at $w_m^t$ and using $\frac{\partial \hat{h}_m}{\partial g_{w_m}} \equiv 1$, we have*

$$\left. \frac{\partial \ell_m}{\partial w_m} \right|_{w_m = w_m^t} = \nabla \hat{h}_m \frac{\partial g_{w_m}}{\partial w_m} = \left. \frac{\partial \mathcal{L}}{\partial \hat{h}_m} \frac{\partial \hat{h}_m}{\partial g_{w_m}} \frac{\partial g_{w_m}}{\partial w_m} \right|_{w_m = w_m^t} = \left. \frac{\partial \mathcal{L}}{\partial w_m} \right|_{w_m = w_m^t}, \tag{8}$$

*where the last equality follows from the chain rule. This concludes the proof.*

**Proposition 2**: Consider a linear function $x \mapsto f_\theta(x) = \theta x$, where $\theta \subset \Theta \subseteq \mathbb{R}^{d_1 \times d_2}$ is the parameter and $x \in \mathbb{R}^{d_2}$. Assume that $g : \mathbb{R}^{d_2} \mapsto \mathbb{R}^{d_1}$ is such a function that $x \mapsto f_\theta(x) + g(x)$ can be equivalently written as $x \mapsto f_{\hat{\theta}}(x)$ for some $\hat{\theta} \in \Theta$. Then, $g$ must be a linear function of $x$ and written as $wx$ for some $w \in \mathbb{R}^{d_2 \times d_1}$.

**Proof 2 (Proof of Proposition 2)** *According to the assumption, there exists $\hat{\theta}$ such that*

$$g(x) = f_{\hat{\theta}}(x) - f_\theta(x) = (\hat{\theta} - \theta)x, \tag{9}$$

*so $g(x)$ can be written as $wx$ with $w = \hat{\theta} - \theta$. This concludes the proof.*

# C  EXPERIMENTAL STUDIES

## C.1  EXPERIMENTAL SETUP

We demonstrate the hyperparameters used in the experiments in Table 5. We use a lower learning rate of 5.00E-6 for ColA (Linear) in some of the GLUE datasets, including MNLI, SST-2, QNLI, QQP, RTE, and Dolly dataset with Lllama-2 base model, because the default learning rate is too large to converge. Due to computational constraints, we did not undertake extensive hyperparameter sweeps. Further tuning of hyperparameters might yield improved results. The auxiliary models of Llama-2 $(Q, V)$ include query and value projection layers, while the auxiliary models of Llama-2 (All) include all projection layers.

Table 5: Hyperparameters used in the experiments.

| Hyperparmeter | FT | PEFT | ColA |
|---|---|---|---|
| Epoch | | 40 | |
| Batch size | | 32 | |
| Optimizer | | AdamW | |
| Weight decay | | 5.00E-04 | |
| Learning rate | 5.00E-06 | 3.00E-04 | |
| Scheduler | | Linear decay | |
| Warm up | | 0.05 | |
| Max sequence length | | 128 | |

## C.2  EXPERIMENTAL RESULTS

The results for GPT-2 and Llama-2 $(Q, V)$ on the CLM task evaluated using the ROUGE (Longest) metric are presented in Table 6 and 7. ColA (Low Rank) demonstrates performance comparable to LoRA with the same number of trainable parameters. Notably, ColA (Linear), despite its larger parameter count, outperforms both FT and LoRA. The sub-optimal performance of FT may be due to a low learning rate, which results in inadequate convergence. Notably, full Fine-Tuning (FT) does not fit in our 48 GB GPU as shown in Table 7.

Table 6: Results of GPT-2 on the Causal Language Modeling (CLM) task with ROUGE (Longest) metric. The gradient of parameters in $\{\cdot\}$ can be stored in low-cost devices. Both parameters and their gradient in $[\cdot]$ can be stored in low-cost devices.

| Method | | Trainable Parameters | Dolly |
|---|---|---|---|
| FT | | 124.4 M (100.0 %) | 15.6 |
| LoRA | | 294.9 K (0.2 %) | 15.6 |
| AdaLoRA | | 2.4 M (1.9 %) | 14.2 |
| IA3 | | 36.9 K (0.03 %) | 14.2 |
| Prompt Tuning | | 15.4 K (0.01 %) | 14.0 |
| Prefix Tuning | | 368.6 K (0.3 %) | 14.5 |
| P-Tuning | | 229.4 K (0.2 %) | 14.6 |
| ColA (Low Rank) | unmerged | {294.9 K (0.2 %)} | 15.5 |
| | merged | [294.9 K (0.2 %)] | 15.6 |
| ColA (Linear) | unmerged | {21.2 M (17.1 %)} | 16.1 |
| | merged | [21.2 M (17.1 %)] | **16.4** |
| ColA (MLP) | unmerged | {8.7 M (7.0 %)} | 15.7 |

Table 7: Results of Llama-2 $(Q, V)$ on the Causal Language Modeling (CLM) task with user collaboration. The gradient of parameters in $\{\cdot\}$ can be stored in low-cost devices. Both parameters and their gradient in $[\cdot]$ can be stored in low-cost devices.

| Method | | Trainable Parameters | Dolly |
|---|---|---|---|
| FT | | 6.7 B (100.0 %) | — |
| LoRA | | 4.2 M (0.06 %) | 18.8 |
| AdaLoRA | | 33.6 M (0.5 %) | 18.9 |
| IA3 | | 393.2 K (0.006 %) | 19.0 |
| Prompt Tuning | | 81.9K (0.001 %) | 18.8 |
| Prefix Tuning | | 5.2 M (0.08 %) | 16.3 |
| P-Tuning | | 1.2 M (0.02 %) | 18.0 |
| ColA (Low Rank) | unmerged | $\{4.2$ M (0.06 %)$\}$ | 19.3 |
| | merged | $[4.2$ M (0.06 %)$]$ | **19.4** |
| ColA (Linear) | unmerged | $\{1.1$ B (15.9 %)$\}$ | 19.1 |
| | merged | $[1.1$ B (15.9 %)$]$ | 19.0 |
| ColA (MLP) | unmerged | $\{103.0$ M (1.5 %)$\}$ | 19.2 |

Table 8: Results of Llama-2 $(Q, V)$ on the Causal Language Modeling task with user collaboration. The gradient of parameters in $\{\cdot\}$ can be stored in low-cost devices. Both parameters and their gradient in $[\cdot]$ can be stored in low-cost devices.

| | Method | | Trainable Parameters | Classification | Information Extraction | Summarization | Brainstorming | Creative Writing | Open Q&A | Closed Q&A | General Q&A | All unmerged | All merged |
|---|---|---|---|---|---|---|---|---|---|---|---|---|---|
| Joint | Low Rank | unmerged | $\{20.0$ M (0.3 %)$\}$ | 24.6 | 16.4 | 16.9 | 18.9 | 15.5 | 20.4 | 17.0 | 19.2 | 19.3 | 19.2 |
| | | merged | $[20.0$ M (0.3 %)$]$ | 25.0 | 16.1 | **18.1** | 19.0 | 15.5 | 20.5 | 17.6 | 19.2 | | **19.4** |
| | Linear | unmerged | $\{6.5$ B (96.1 %)$\}$ | **25.7** | 15.3 | 16.1 | **20.1** | 14.9 | 20.3 | 16.3 | 19.2 | 19.1 | 19.2 |
| | | merged | $[6.5$ B (96.1 %)$]$ | 25.4 | 14.1 | 16.3 | 19.5 | 15.1 | **20.7** | 16.1 | 18.9 | | 19.0 |
| | MLP | unmerged | $\{502.7$ M (7.5 %)$\}$ | 24.0 | 16.4 | 17.1 | 19.6 | **15.7** | 20.6 | 16.9 | 18.5 | 19.2 | — |
| Alone | Low Rank | | $8 \times \{20.0$ M$\}$ | 24.5 | **16.5** | 16.4 | 19.1 | 15.2 | 20.4 | **19.3** | 18.8 | 19.4 | 17.4 |
| | Low Rank-MLP | | $4 \times \{20.0$ M$\}, 4 \times \{502.7$ M$\}$ | 23.2 | 14.2 | 17.2 | 19.0 | 14.3 | 20.5 | 19.2 | **19.4** | 19.2 | — |
| Collaboration | Low Rank | | $8 \times [20.0$ M$]$ | 23.8 | 14.7 | 16.0 | 19.5 | 15.8 | 20.4 | 16.2 | 19.0 | 18.8 | |

## C.3 LEARNING FROM SCRATCH

Recall that ColA (merged) can reduce the cost of full fine-tuning by offloading the computation of the gradient of parameters to other devices. It indicates that our method can achieve the performance of full parameter training from scratch while reducing the computation space bottleneck. To corroborate this, we trained the MNIST and CIFAR10 datasets on Linear, MLP, and CNN models from scratch. We train these models for 400 epochs with Stochastic Gradient Descent (SGD) and cosine annealing learning rate. The results demonstrate that LoRA yields suboptimal results due to low-rank approximation while our method can achieve the results of full-fine tuning because we can train the model without any approximation. Furthermore, MLP auxiliary models may also outperform full fine-tuning due to over-parameterization.

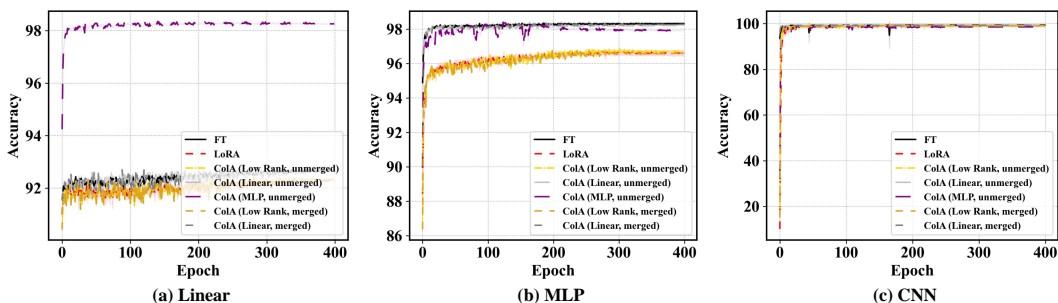

Figure 2: Learning curves of (a) Linear (b) MLP and (c) CNN with the MNIST dataset of IC task and Accuracy metric.

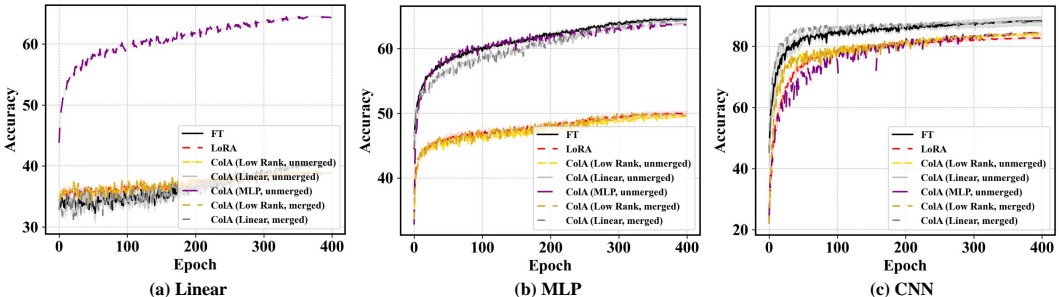

**(a) Linear**        **(b) MLP**        **(c) CNN**

Figure 3: Learning curves of (a) Linear (b) MLP and (c) CNN with the CIFAR10 dataset of IC task and Accuracy metric.

Table 9: Results of learning Linear, MLP, and CNN from scratch on the Image Classification (IC) task with Accuracy metric. The gradient of parameters in $\{\cdot\}$ can be stored in low-cost devices. Both parameters and their gradient in $[\cdot]$ can be stored in low-cost devices.

| Model | Method | | Trainable Parameters | MNIST | CIFAR10 |
|---|---|---|---|---|---|
| Linear | FT | | 7.9 K (100.0 %) | 92.8 | 40.6 |
| | LoRA | | 6.4k (80.9 %) | 92.4 | 39.0 |
| | ColA (Low Rank) | unmerged | $\{6.4k\ (80.9\ \%)\}$ | 92.4 | 39.2 |
| | | merged | $[6.4k\ (80.9\ \%)]$ | 92.4 | 39.2 |
| | ColA (Linear) | unmerged | $\{7.9\ K\ (100.0\ \%)\}$ | 92.7 | 40.7 |
| | | merged | $[7.9\ K\ (100.0\ \%)]$ | 92.7 | 40.7 |
| | ColA (MLP) | unmerged | $\{136.1K\ (1733.4\ \%)\}$ | **98.4** | **64.7** |
| MLP | FT | | 136.1 K (100.0 %) | 98.4 | **64.7** |
| | LoRA | | 12.5 K (9.2 %) | 96.8 | 50.1 |
| | ColA (Low Rank) | unmerged | $\{12.5\ K\ (9.2\ \%)\}$ | 96.8 | 49.6 |
| | | merged | $[12.5\ K\ (9.2\ \%)]$ | 96.8 | 50.2 |
| | ColA (Linear) | unmerged | $\{136.1\ K\ (100.0\ \%)\}$ | 98.3 | 64.1 |
| | | merged | $[136.1\ K\ (100.0\ \%)]$ | 98.3 | **64.6** |
| | ColA (MLP) | unmerged | $\{350.2\ K\ (257.4\ \%)\}$ | **98.4** | 63.9 |
| CNN | FT | | 155.5 K (100.0 %) | 99.4 | 88.3 |
| | LoRA | | 44.2 K (2.8 %) | 99.1 | 82.9 |
| | ColA (Low Rank) | unmerged | $\{44.2\ K\ (2.8\ \%)\}$ | 99.2 | 84.0 |
| | | merged | $[44.2\ K\ (2.8\ \%)]$ | 99.3 | 84.4 |
| | ColA (Linear) | unmerged | $\{155.5\ K\ (100.0\ \%)\}$ | 99.4 | **88.3** |
| | | merged | $[155.5\ K\ (100.0\ \%)]$ | **99.5** | 88.1 |
| | ColA (MLP) | unmerged | $\{538.1\ K\ (34.6\ \%)\}$ | 98.9 | 84.5 |

## C.4 ADAPTATION INTERVAL

We conduct ablation studies regarding the adaptation interval $I$. For these experiments, we use a batch size of $B = 8$ and ColA (unmerged). By increasing the adaptation interval, such as $I = 4$, we can effectively increase the batch size from $B$ to $B \times I$. In these experiments, we maintain the same number of training iterations $T$ for different adaptation intervals. Therefore, experiments with $I = 8$ will update the auxiliary models at one-eighth the frequency of those experiments with $I = 1$. Furthermore, the results demonstrate that by tuning a proper adaptation interval, we can use less communication cost to achieve satisfactory performance. In particular, with a large effective batch size, we can estimate the gradient of parameters more accurately and potentially speed up the model convergence. This extension becomes especially valuable in situations demanding extensive computational space for computing the gradient of hidden representations for numerous users of FTaaS.

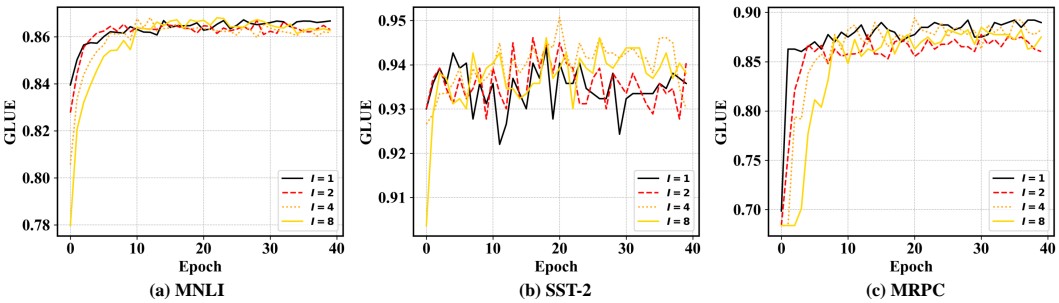

Figure 4: Ablation studies of adaptation interval $I$ on (a) MNLI (b) SST-2, and (c) MRPC datasets of SC task and GLUE metric.

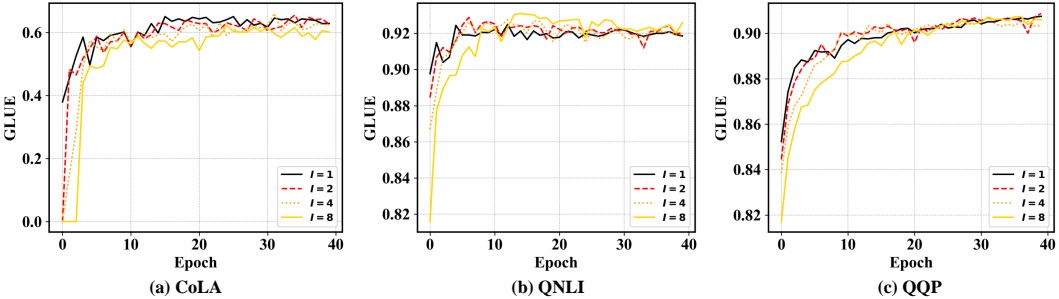

Figure 5: Ablation studies of adaptation interval $I$ on (a) CoLA (b) QNLI, and (c) QQP datasets of SC task and GLUE metric.

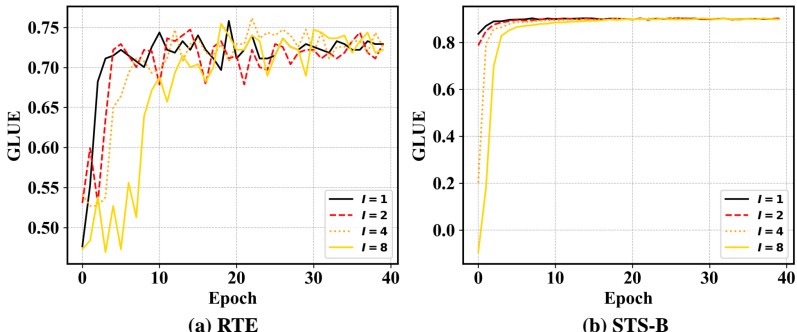

Figure 6: Ablation studies of adaptation interval $I$ on (a) RTE and (b) STS-B datasets of SC task and GLUE metric.

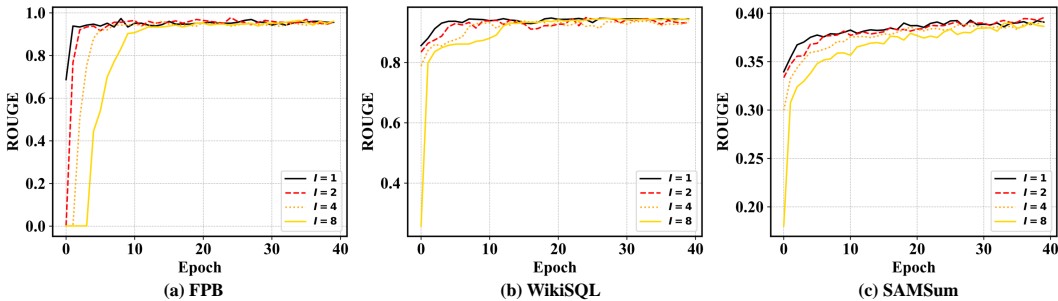

Figure 7: Ablation studies of adaptation interval $I$ on (a) FPB (b) WikiSQL, and (c) SAMSum datasets of S2S task and ROUGE (Longest) metric.

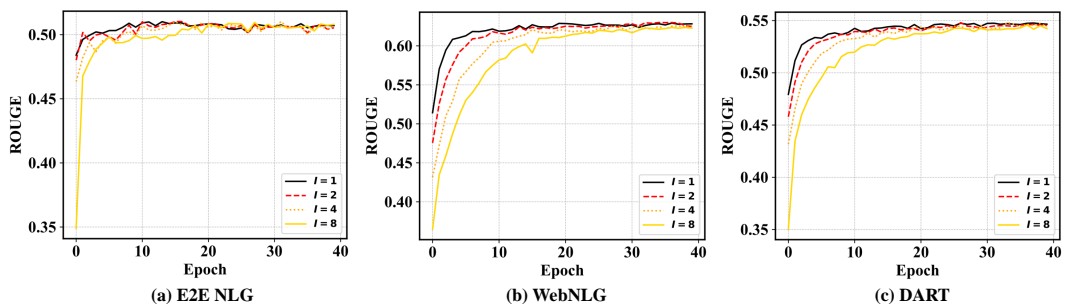

Figure 8: Ablation studies of adaptation interval $I$ on (a) E2E NLG (b) WebNLG, and (c) DART datasets of S2S task and ROUGE (Longest) metric.

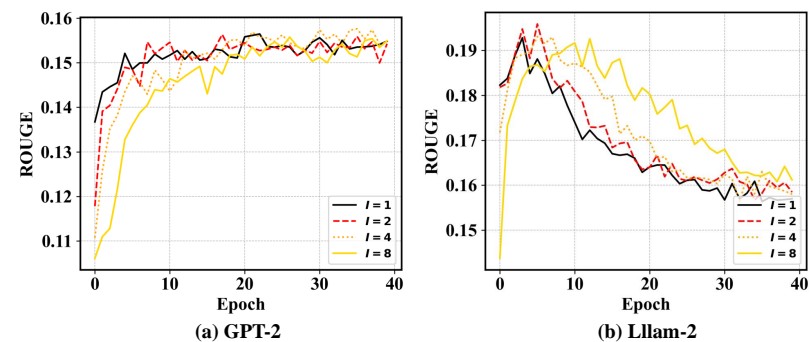

Figure 9: Ablation studies of adaptation interval $I$ of (a) GPT-2 and (b) Llama-2 $(Q, V)$ on Dolly dataset of CLM task and ROUGE (Longest) metric.

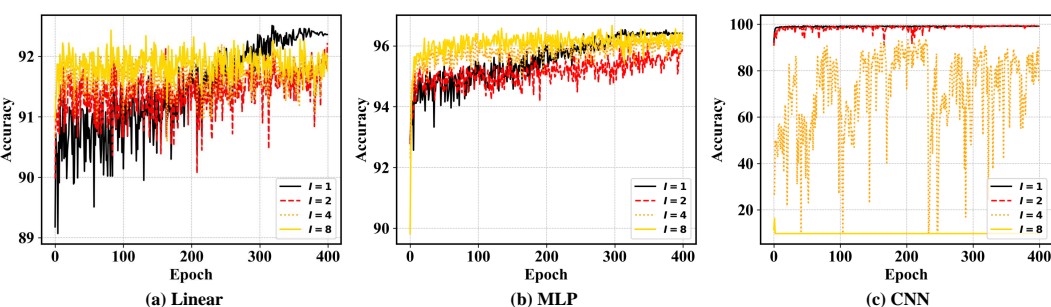

Figure 10: Ablation studies of adaptation interval $I$ of (a) Linear, (b) MLP, and (c) CNN on MNIST dataset of IC task and Accuracy metric.

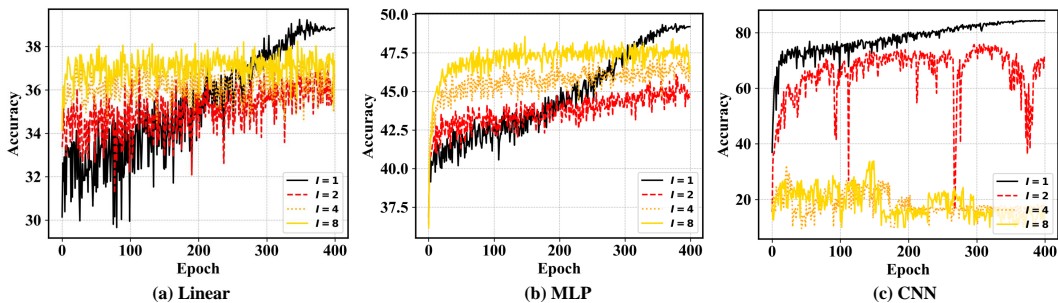

**(a) Linear**       **(b) MLP**       **(c) CNN**

Figure 11: Ablation studies of adaptation interval $I$ of (a) Linear, (b) MLP, and (c) CNN on CI-FAR10 dataset of IC task and Accuracy metric.

## C.5 COMPUTATION EVALUATION

We conduct a comprehensive quantitative evaluation of computational costs for all feasible experiments on real devices. We use an NVIDIA RTX A6000 (48 GB) as the primary host for the base model, with computation offloaded to a secondary device, either a CPU (Intel Xeon Gold 5320) with 944 GB RAM or another A6000. We carry out experiments by executing 10 training iterations with batch sizes of 1, 8, and 32, during which we measure computation space and run time.

The memory consumption of the primary device hosting the base model is noted as 'Memory (Base)'. To accurately evaluate the computation space for the gradient of auxiliary parameters, we measure the difference in memory consumption between two specific scenarios: one where computation is offloaded to a CPU, and another where it remains on the primary device. This difference in memory usage between these two cases is denoted as 'Memory (Offload)'. When the total memory usage exceeds 48 GB, we report the remaining memory on the primary GPU for 'Memory (Offload)'.

We also track the run time of the forward and backward pass of the base model, including the time taken to transfer adaptation data to another device, labeled as 'Offload to CPU (Base)' and 'Offload to GPU (Base)'. Additionally, we measure the run time for updating the auxiliary models on the secondary device, reporting the averaged run time across all adapters, as these models can be updated in parallel. In certain cases presented in Table 18, when the secondary A6000 GPU lacks the capacity to handle the offloaded memory, we do not report the run time.

The results, outlined in Tables 10 to 18, demonstrate a significant reduction in computation space bottleneck during the back-propagation of the base model through our method, while also indicating that run time can be decreased by offloading to an additional GPU. More specifically,

- Our method, ColA (Low Rank), which matches LoRA exactly, consistently uses less computation space than LoRA during back-propagation of the base model.

- Despite the limited size of auxiliary models only providing a small advantage over LoRA (2-4 GB as shown in Table 13 and 14), our ColA (Linear, merged) can train the network without low-rank approximation and significantly reduce the computation space bottleneck. For example, as shown in Table 13, ColA (Linear, merged, batch size=8) requires 20.8 GB, compared to LoRA's 21.8 GB. Similarly, Table 14 shows that ColA (Linear, merged, batch size=32) uses 42.8 GB, while LoRA exceeds 48 GB and does not fit on our device. Notably, our method also requires less computation space than direct training on the base model. For instance, full fine-tuning in Table 14 does not fit in 48 GB device even at a batch size of one.

- As demonstrated from Tables 16 to 18, ColA (merged) consumes the same amount of memory for a given batch size, regardless of the size of auxiliary models and the number of users, because the computation of all auxiliary models has been completely offloaded to a separate device.

- The results demonstrate that the run time scales with batch size when we offload the computation to a CPU. However, offloading computation to an additional GPU significantly reduces run time. It is foreseeable as the communication of tensors among GPUs is much faster than it is between GPU and CPU. A pivotal aspect of our method is that the computation of the gradient of model parameters can be *decoupled* from the classical back-propagation and *distributed* across multiple smaller devices with lower memory capacity. This is particularly desirable considering the affordability and availability of consumer-grade GPUs with smaller memory capacities, such as the 3090, in contrast to professional-grade GPUs like the H100.

Table 10: Computation evaluation of SC task with RoBERTa (base) model and GLUE (CoLA) dataset, where the number of auxiliary models $M = 26$.

| Batch Size | Method | | Trainable Parameters | Memory | | Run Time (s) | | | |
|---|---|---|---|---|---|---|---|---|---|
| | | | | | | Offload to CPU | | Offload to GPU | |
| | | | | Base | Offload | Base | Offload | Base | Offload |
| 1 | FT | | 125.2 M (100.0 %) | 3.6 GB | | 0.037 | | | |
| | LoRA | | 887.0 K (0.7 %) | 1.1 GB | | 0.037 | | | |
| | AdaLoRA | | 11.3 M (9.0 %) | 1.3 GB | | 0.093 | | | |
| | IA3 | | 629.0 K (0.5 %) | 1.1 GB | | 0.030 | | | |
| | Prompt Tuning | | 607.5 K (0.5 %) | 1.1 GB | | 0.029 | | | |
| | Prefix Tuning | | 960.8 K (0.8 %) | 1.1 GB | | 0.028 | | | |
| | P-Tuning | | 821.5 K (0.7 M %) | 1.1 GB | | 0.028 | | | |
| | ColA (Low Rank) | unmerged | {887.0 K (0.7 %)} | 1.1 GB | 3.0 MB | 0.091 | 0.002 | 0.034 | 0.001 |
| | | merged | [887.0 K (0.7 %)] | 1.1 GB | 4.0 MB | 0.081 | 0.002 | 0.066 | 0.001 |
| | ColA (Linear) | unmerged | {14.7 M (11.8 %)} | 1.2 GB | 120.0 MB | 0.055 | 0.003 | 0.035 | 0.001 |
| | | merged | [14.7 M (11.8 %)] | **1.1 GB** | 180.0 MB | 0.071 | 0.002 | 0.073 | 0.001 |
| | ColA (MLP) | unmerged | {8.5 M (6.7 %)} | 1.1 GB | 60.0 MB | 0.063 | 0.006 | 0.039 | 0.002 |
| 8 | FT | | 125.2 M (100.0 %) | 3.6 GB | | 0.040 | | | |
| | LoRA | | 887.0 K (0.7 %) | 1.6 GB | | 0.043 | | | |
| | AdaLoRA | | 11.3 M (9.0 %) | 2.1 GB | | 0.108 | | | |
| | IA3 | | 629.0 K (0.5 %) | 1.7 GB | | 0.032 | | | |
| | Prompt Tuning | | 607.5 K (0.5 %) | 1.8 GB | | 0.026 | | | |
| | Prefix Tuning | | 960.8 K (0.8 %) | 1.6 GB | | 0.034 | | | |
| | P-Tuning | | 821.5 K (0.7 M %) | 1.8 GB | | 0.026 | | | |
| | ColA (Low Rank) | unmerged | {887.0 K (0.7 %)} | 1.6 GB | 12.8 MB | 0.166 | 0.003 | 0.037 | 0.001 |
| | | merged | [887.0 K (0.7 %)] | 1.6 GB | 5.2 MB | 0.146 | 0.003 | 0.077 | 0.001 |
| | ColA (Linear) | unmerged | {14.7 M (11.8 %)} | 1.7 GB | 130.8 MB | 0.132 | 0.007 | 0.035 | 0.001 |
| | | merged | [14.7 M (11.8 %)] | **1.6 GB** | 212.0 MB | 0.159 | 0.006 | 0.062 | 0.001 |
| | ColA (MLP) | unmerged | {8.5 M (6.7 %)} | 1.7 GB | 70.8 MB | 0.157 | 0.009 | 0.041 | 0.002 |
| 32 | FT | | 125.2 M (100.0 %) | 5.5 GB | | 0.093 | | | |
| | LoRA | | 887.0 K (0.7 %) | 3.3 GB | | 0.033 | | | |
| | AdaLoRA | | 11.3 M (9.0 %) | 4.4 GB | | 0.099 | | | |
| | IA3 | | 629.0 K (0.5 %) | 3.7 GB | | 0.028 | | | |
| | Prompt Tuning | | 607.5 K (0.5 %) | 3.8 GB | | 0.021 | | | |
| | Prefix Tuning | | 960.8 K (0.8 %) | 3.4 GB | | 0.024 | | | |
| | P-Tuning | | 821.5 K (0.7 M %) | 3.8 GB | | 0.029 | | | |
| | ColA (Low Rank) | unmerged | {887.0 K (0.7 %)} | 3.2 GB | 4.0 MB | 0.273 | 0.005 | 0.052 | 0.003 |
| | | merged | [887.0 K (0.7 %)] | 3.2 GB | 10.0 MB | 0.264 | 0.004 | 0.081 | 0.004 |
| | ColA (Linear) | unmerged | {14.7 M (11.8 %)} | 3.3 GB | 132.0 MB | 0.300 | 0.012 | 0.047 | 0.003 |
| | | merged | [14.7 M (11.8 %)] | **3.2 GB** | 200.0 MB | 0.281 | 0.009 | 0.078 | 0.004 |
| | ColA (MLP) | unmerged | {8.5 M (6.7 %)} | 3.4 GB | 62.0 MB | 0.273 | 0.013 | 0.045 | 0.003 |

Table 11: Computation evaluation of S2S task with BART (base) model and FPB dataset, where the number of auxiliary models $M = 36$

| Batch Size | Method | | Trainable Parameters | Memory | | Run Time (s) | | | |
|---|---|---|---|---|---|---|---|---|---|
| | | | | | | Offload to CPU | | Offload to GPU | |
| | | | | Base | Offload | Base | Offload | Base | Offload |
| 1 | FT | | 139.4 M (100.0 %) | 3.9 GB | | 0.054 | | | |
| | LoRA | | 442.4 K (0.3 %) | 1.1 GB | | 0.049 | | | |
| | AdaLoRA | | 13.0 M (9.3 %) | 1.4 GB | | 0.116 | | | |
| | IA3 | | 36.9 K (0.03 %) | 1.2 GB | | 0.039 | | | |
| | Prompt Tuning | | 30.7 K (0.02 %) | 1.1 GB | | 0.028 | | | |
| | Prefix Tuning | | 184.3 K (0.1 %) | 1.1 GB | | 0.028 | | | |
| | P-Tuning | | 244.7 K (0.2 %) | 1.1 GB | | 0.028 | | | |
| | ColA (Low Rank) | unmerged | {442.4 K (0.3 %)} | 1.1 GB | 2.0 MB | 0.099 | 0.002 | 0.054 | 0.001 |
| | | merged | [442.4 K (0.3 %)] | 1.1 GB | 6.0 MB | 0.095 | 0.001 | 0.053 | 0.001 |
| | ColA (Linear) | unmerged | {21.2 M (15.2 %)} | 1.2 GB | 180.0 MB | 0.078 | 0.004 | 0.045 | 0.001 |
| | | merged | [21.2 M (15.2 %)] | **1.1 GB** | 260.0 MB | 0.137 | 0.003 | 0.078 | 0.001 |
| | ColA (MLP) | unmerged | {11.8 M (8.5 %)} | 1.2 GB | 90.0 MB | 0.091 | 0.006 | 0.068 | 0.002 |
| 8 | FT | | 139.4 M (100.0 %) | 3.9 GB | | 0.037 | | | |
| | LoRA | | 442.4 K (0.3 %) | 1.5 GB | | 0.044 | | | |
| | AdaLoRA | | 13.0 M (9.3 %) | 1.9 GB | | 0.120 | | | |
| | IA3 | | 36.9 K (0.03 %) | 1.5 GB | | 0.037 | | | |
| | Prompt Tuning | | 30.7 K (0.02 %) | 1.5 GB | | 0.029 | | | |
| | Prefix Tuning | | 184.3 K (0.1 %) | 1.2 GB | | 0.030 | | | |
| | P-Tuning | | 244.7 K (0.2 %) | 1.5 GB | | 0.034 | | | |
| | ColA (Low Rank) | unmerged | {442.4 K (0.3 %)} | 1.4 GB | 4.0 MB | 0.155 | 0.003 | 0.058 | 0.001 |
| | | merged | [442.4 K (0.3 %)] | 1.4 GB | 6.0 MB | 0.118 | 0.002 | 0.061 | 0.001 |
| | ColA (Linear) | unmerged | {21.2 M (15.2 %)} | 1.6 GB | 184.0 MB | 0.142 | 0.005 | 0.047 | 0.001 |
| | | merged | [21.2 M (15.2 %)] | **1.4 GB** | 304.0 MB | 0.132 | 0.004 | 0.092 | 0.001 |
| | ColA (MLP) | unmerged | {11.8 M (8.5 %)} | 1.5 GB | 90.0 MB | 0.152 | 0.007 | 0.057 | 0.002 |
| 32 | FT | | 139.4 M (100.0 %) | 4.5 GB | | 0.069 | | | |
| | LoRA | | 442.4 K (0.3 %) | 2.6 GB | | 0.055 | | | |
| | AdaLoRA | | 13.0 M (9.3 %) | 3.3 GB | | 0.129 | | | |
| | IA3 | | 36.9 K (0.03 %) | 2.8 GB | | 0.034 | | | |
| | Prompt Tuning | | 30.7 K (0.02 %) | 2.8 GB | | 0.033 | | | |
| | Prefix Tuning | | 184.3 K (0.1 %) | 1.5 GB | | 0.024 | | | |
| | P-Tuning | | 244.7 K (0.2 %) | 2.8 GB | | 0.029 | | | |
| | ColA (Low Rank) | unmerged | {442.4 K (0.3 %)} | 2.5 GB | 4.0 MB | 0.255 | 0.003 | 0.059 | 0.001 |
| | | merged | [442.4 K (0.3 %)] | 2.5 GB | 6.0 MB | 0.256 | 0.003 | 0.066 | 0.001 |
| | ColA (Linear) | unmerged | {21.2 M (15.2 %)} | 2.6 GB | 184.0 MB | 0.258 | 0.007 | 0.051 | 0.001 |
| | | merged | [21.2 M (15.2 %)] | **2.5 GB** | 272.0 MB | 0.271 | 0.007 | 0.092 | 0.002 |
| | ColA (MLP) | unmerged | {11.8 M (8.5 %)} | 2.6 GB | 90.0 MB | 0.234 | 0.009 | 0.065 | 0.002 |

Table 12: Computation evaluation of CLM task with GPT-2 model and Dolly dataset, where the number of auxiliary models $M = 12$

| Batch Size | Method | | Trainable Parameters | Memory | | Run Time (s) | | | |
|---|---|---|---|---|---|---|---|---|---|
| | | | | | | Offload to CPU | | Offload to GPU | |
| | | | | Base | Offload | Base | Offload | Base | Offload |
| 1 | FT | | 124.4 M (100.0 %) | 3.6 GB | | 0.031 | | | |
| | LoRA | | 294.9 K (0.2 %) | 1.3 GB | | 0.027 | | | |
| | AdaLoRA | | 2.4 M (1.9 %) | 1.3 GB | | 0.032 | | | |
| | IA3 | | 36.9 K (0.03 %) | 1.3 GB | | 0.023 | | | |
| | Prompt Tuning | | 15.4 K (0.01 %) | 1.3 GB | | 0.021 | | | |
| | Prefix Tuning | | 368.6 K (0.3 %) | 1.3 GB | | 0.023 | | | |
| | P-Tuning | | 229.4 K (0.2 %) | 1.3 GB | | 0.028 | | | |
| | ColA (Low Rank) | unmerged | {294.9 K (0.2 %)} | 1.3 GB | 2.0 MB | 0.063 | 0.002 | 0.033 | 0.001 |
| | | merged | [294.9 K (0.2 %)] | 1.3 GB | 4.0 MB | 0.078 | 0.002 | 0.047 | 0.001 |
| | ColA (Linear) | unmerged | {21.2 M (17.1 %)} | 1.3 GB | 232.0 MB | 0.066 | 0.005 | 0.034 | 0.001 |
| | | merged | [21.2 M (17.1 %)] | **1.3 GB** | 312.0 MB | 0.066 | 0.004 | 0.042 | 0.001 |
| | ColA (MLP) | unmerged | {8.7 M (7.0 %)} | 1.3 GB | 64.0 MB | 0.080 | 0.007 | 0.035 | 0.002 |
| 8 | FT | | 124.4 M (100.0 %) | 4.4 GB | | 0.029 | | | |
| | LoRA | | 294.9 K (0.2 %) | 3.1 GB | | 0.027 | | | |
| | AdaLoRA | | 2.4 M (1.9 %) | 3.2 GB | | 0.033 | | | |
| | IA3 | | 36.9 K (0.03 %) | 3.2 GB | | 0.027 | | | |
| | Prompt Tuning | | 15.4 K (0.01 %) | 3.4 GB | | 0.025 | | | |
| | Prefix Tuning | | 368.6 K (0.3 %) | 3.0 GB | | 0.024 | | | |
| | P-Tuning | | 229.4 K (0.2 %) | 3.4 GB | | 0.026 | | | |
| | ColA (Low Rank) | unmerged | {294.9 K (0.2 %)} | 3.1 GB | 2.0 MB | 0.131 | 0.003 | 0.043 | 0.001 |
| | | merged | [294.9 K (0.2 %)] | 3.1 GB | 2.0 MB | 0.140 | 0.004 | 0.052 | 0.001 |
| | ColA (Linear) | unmerged | {21.2 M (17.1 %)} | 3.2 GB | 196.0 MB | 0.158 | 0.012 | 0.035 | 0.001 |
| | | merged | [21.2 M (17.1 %)] | **3.1 GB** | 276.0 MB | 0.174 | 0.010 | 0.062 | 0.001 |
| | ColA (MLP) | unmerged | {8.7 M (7.0 %)} | 3.1 GB | 12.0 MB | 0.146 | 0.015 | 0.038 | 0.002 |
| 32 | FT | | 124.4 M (100.0 %) | 10.7 GB | | 0.150 | | | |
| | LoRA | | 294.9 K (0.2 %) | 9.2 GB | | 0.050 | | | |
| | AdaLoRA | | 2.4 M (1.9 %) | 9.2 GB | | 0.057 | | | |
| | IA3 | | 36.9 K (0.03 %) | 9.6 GB | | 0.046 | | | |
| | Prompt Tuning | | 15.4 K (0.01 %) | 10.5 GB | | 0.055 | | | |
| | Prefix Tuning | | 368.6 K (0.3 %) | 8.8 GB | | 0.048 | | | |
| | P-Tuning | | 229.4 K (0.2 %) | 10.5 GB | | 0.056 | | | |
| | ColA (Low Rank) | unmerged | {294.9 K (0.2 %)} | 9.0 GB | 4.0 MB | 0.998 | 0.015 | 0.080 | 0.007 |
| | | merged | [294.9 K (0.2 %)] | 9.0 GB | 16.0 MB | 0.955 | 0.025 | 0.083 | 0.009 |
| | ColA (Linear) | unmerged | {21.2 M (17.1 %)} | 9.1 GB | 0B | 1.012 | 0.027 | 0.089 | 0.007 |
| | | merged | [21.2 M (17.1 %)] | **9.0 GB** | 92.0 MB | 0.959 | 0.034 | 0.081 | 0.008 |
| | ColA (MLP) | unmerged | {8.7 M (7.0 %)} | 9.1 GB | 12.0 MB | 1.026 | 0.032 | 0.092 | 0.008 |

Table 13: Computation evaluation of CLM task with Llama-2 ($Q$, $V$) model and Dolly dataset, where the number of auxiliary models $M = 64$. The auxiliary models of Llama-2 ($Q$, $V$) include query and value projection layers.

| Batch Size | Method | | Trainable Parameters | Memory | | Run Time (s) Offload to CPU | | Offload to GPU | |
|---|---|---|---|---|---|---|---|---|---|
| | | | | Base | Offload | Base | Offload | Base | Offload |
| 1 | FT | | 6.7 B (100.0 %) | — | | — | | | |
| | LoRA | | 4.2 M (0.06 %) | 14.2 GB | | 0.105 | | | |
| | AdaLoRA | | 33.6 M (0.5 %) | 14.8 GB | | 0.146 | | | |
| | IA3 | | 393.2 K (0.006 %) | 14.2 GB | | 0.107 | | | |
| | Prompt Tuning | | 81.9K (0.001 %) | 14.0 GB | | 0.082 | | | |
| | Prefix Tuning | | 5.2 M (0.08 %) | 14.1 GB | | 0.086 | | | |
| | P-Tuning | | 1.2 M (0.02 %) | 14.2 GB | | 0.084 | | | |
| | ColA (Low Rank) | unmerged | {4.2 M (0.06 %)} | 14.0 GB | 32.0 MB | 0.274 | 0.003 | 0.135 | 0.001 |
| | | merged | [4.2 M (0.06 %)] | 14.0 GB | 120.0 MB | 3.097 | 0.002 | — | |
| | ColA (Linear) | unmerged | {1.1 B (15.9 %} | 18.0 GB | 8.2 GB | 0.213 | 0.095 | 0.153 | 0.001 |
| | | merged | [1.1 B (15.9 %)] | **14.0 GB** | 12.2 GB | 2.675 | 0.040 | — | |
| | ColA (MLP) | unmerged | {103.0 M (1.5 %)} | 14.4 GB | 790.0 MB | 0.248 | 0.007 | 0.157 | 0.002 |
| 8 | FT | | 6.7 B (100.0 %) | — | | — | | | |
| | LoRA | | 4.2 M (0.06 %) | 21.8 GB | | 0.197 | | | |
| | AdaLoRA | | 33.6 M (0.5 %) | 22.4 GB | | 0.204 | | | |
| | IA3 | | 393.2 K (0.006 %) | 22.3 GB | | 0.190 | | | |
| | Prompt Tuning | | 81.9K (0.001 %) | 22.3 GB | | 0.207 | | | |
| | Prefix Tuning | | 5.2 M (0.08 %) | 21.0 GB | | 0.177 | | | |
| | P-Tuning | | 1.2 M (0.02 %) | 22.3 GB | | 0.205 | | | |
| | ColA (Low Rank) | unmerged | {4.2 M (0.06 %)} | 20.6 GB | 32.0 MB | 0.607 | 0.004 | 0.277 | 0.002 |
| | | merged | [4.2 M (0.06 %)] | 20.8 GB | 68.0 MB | 2.294 | 0.004 | — | |
| | ColA (Linear) | unmerged | {1.1 B (15.9 %} | 24.6 GB | 8.0 GB | 0.683 | 0.113 | 0.360 | 0.005 |
| | | merged | [1.1 B (15.9 %)] | **20.8 GB** | 12.0 GB | 1.598 | 0.052 | — | |
| | ColA (MLP) | unmerged | {103.0 M (1.5 %)} | 21.3 GB | 798.0 MB | 0.640 | 0.015 | 0.279 | 0.002 |
| 32 | FT | | 6.7 B (100.0 %) | — | | — | | | |
| | LoRA | | 4.2 M (0.06 %) | 46.6 GB | | 0.762 | | | |
| | AdaLoRA | | 33.6 M (0.5 %) | 47.3 GB | | 0.856 | | | |
| | IA3 | | 393.2 K (0.006 %) | — | | — | | | |
| | Prompt Tuning | | 81.9K (0.001 %) | — | | — | | | |
| | Prefix Tuning | | 5.2 M (0.08 %) | 43.1 GB | | 0.686 | | | |
| | P-Tuning | | 1.2 M (0.02 %) | — | | — | | | |
| | ColA (Low Rank) | unmerged | {4.2 M (0.06 %)} | 42.7 GB | 32.0 MB | 8.893 | 0.063 | — | |
| | | merged | [4.2 M (0.06 %)] | 42.7 GB | 52.0 MB | 8.069 | 0.081 | — | |
| | ColA (Linear) | unmerged | {1.1 B (15.9 %} | 46.7 GB | > 1.3 GB | 9.260 | 0.244 | — | |
| | | merged | [1.1 B (15.9 %)] | **42.7 GB** | > 5.3 GB | 7.392 | 0.233 | — | |
| | ColA (MLP) | unmerged | {103.0 M (1.5 %)} | 43.4 GB | 842.0 MB | 8.789 | 0.075 | — | |

Table 14: Computation evaluation of CLM task with Llama-2 (All) model and Dolly dataset, where the number of auxiliary models $M = 228$. The auxiliary models of Llama-2 (All) include all projection layers.

| Batch Size | Method | | Trainable Parameters | Memory | | Run Time (s) | | | |
|---|---|---|---|---|---|---|---|---|---|
| | | | | | | Offload to CPU | | Offload to GPU | |
| | | | | Base | Offload | Base | Offload | Base | Offload |
| 1 | FT | | 6.7 B (100.0 %) | — | | — | | | |
| | LoRA | | 20.0 M (0.3 %) | 15.0 GB | | 0.130 | | | |
| | AdaLoRA | | 160.0 M (2.4 %) | 17.7 GB | | 0.285 | | | |
| | ColA (Low Rank) | unmerged | {20.0 M (0.3 %)} | 14.1 GB | 160.0 MB | 0.618 | 0.002 | 0.395 | 0.002 |
| | | merged | [20.0 M (0.3 %)] | 14.1 GB | 302.0 MB | 14.938 | 0.002 | — | |
| | ColA (Linear) | unmerged | {6.5 B (96.1 %)} | 38.2 GB | > 9.8 GB | 0.579 | 0.163 | — | |
| | | merged | [6.5 B (96.1 %)] | **14.1 GB** | > 33.9 GB | 13.311 | 0.064 | — | |
| | ColA (MLP) | unmerged | {502.7 M (7.5 %)} | 16.0 GB | 4.0 GB | 0.529 | 0.009 | 0.217 | 0.002 |
| 8 | FT | | 6.7 B (100.0 %) | — | | — | | | |
| | LoRA | | 20.0 M (0.3 %) | 25.6 GB | | 0.257 | | | |
| | AdaLoRA | | 160.0 M (2.4 %) | 28.2 GB | | 0.359 | | | |
| | ColA (Low Rank) | unmerged | {20.0 M (0.3 %)} | 20.8 GB | 160.0 MB | 4.308 | 0.007 | — | |
| | | merged | [20.0 M (0.3 %)] | 20.9 GB | 160.0 MB | 13.768 | 0.008 | — | |
| | ColA (Linear) | unmerged | {6.5 B (96.1 %)} | 44.9 GB | > 3.1 GB | 4.708 | 0.189 | — | |
| | | merged | [6.5 B (96.1 %)] | **20.9 GB** | > 27.1 GB | 13.217 | 0.107 | — | |
| | ColA (MLP) | unmerged | {502.7 M (7.5 %)} | 23.2 GB | 3.9 GB | 4.412 | 0.019 | 0.847 | 0.003 |
| 32 | FT | | 6.7 B (100.0 %) | — | | — | | | |
| | LoRA | | 20.0 M (0.3 %) | — | | — | | | |
| | AdaLoRA | | 160.0 M (2.4 %) | — | | — | | | |
| | ColA (Low Rank) | unmerged | {20.0 M (0.3 %)} | 43.2 GB | 160.0 MB | 41.026 | 0.083 | — | |
| | | merged | [20.0 M (0.3 %)] | 42.8 GB | 320.0 MB | 38.714 | 0.103 | — | |
| | ColA (Linear) | unmerged | {6.5 B (96.1 %)} | — | | — | | — | |
| | | merged | [6.5 B (96.1 %)] | 42.8 GB | > 5.2 GB | 36.571 | 0.352 | — | |
| | ColA (MLP) | unmerged | {502.7 M (7.5 %)} | 45.9 GB | > 2.1 GB | 40.676 | 0.099 | — | |

## C.6 LEARNING CURVES

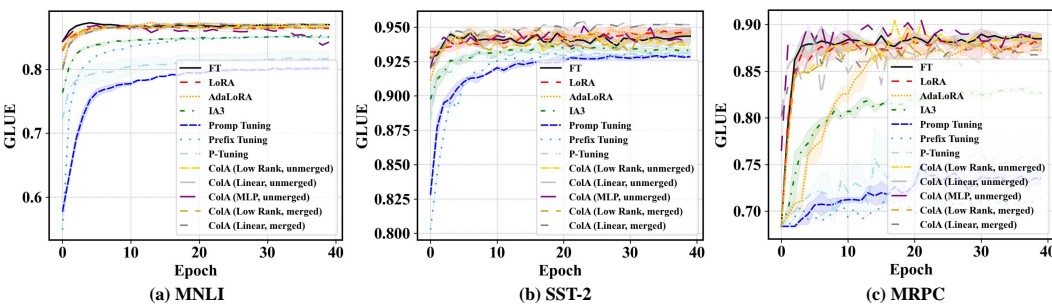

Figure 12: Learning curves of (a) MNLI (b) SST-2, and (c) MRPC datasets of SC task with and GLUE metric.

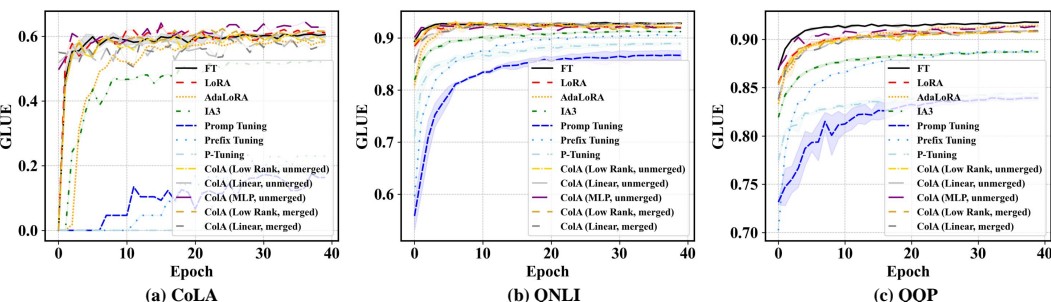

Figure 13: Learning curves of (a) CoLA (b) QNLI, and (c) QQP datasets of SC task and GLUE metric.

Table 15: Computation evaluation of IC task with Linear, MLP, CNN models and MNIST dataset.

| Model | Batch Size | Method | | Trainable Parameters | Memory | | Run Time (s) Offload to CPU | | Offload to GPU | |
|---|---|---|---|---|---|---|---|---|---|---|
| | | | | | Base | Offload | Base | Offload | Base | Offload |
| Linear, $M=1$ | 1 | FT | | 7.9 K (100.0 %) | 545.8 MB | | 0.001 | | | |
| | | LoRA | | 6.4k (80.9 %) | 545.8 MB | | 0.001 | | | |
| | | ColA (Low Rank) | unmerged | {6.4k (80.9 %)} | 537.8 MB | 8.0 MB | 0.001 | 0.001 | 0.001 | 0.001 |
| | | | merged | [6.4k (80.9 %)] | 537.8 MB | 8.0 MB | 0.001 | 0.001 | 0.001 | 0.001 |
| | | ColA (Linear) | unmerged | {7.9 K (100.0 %)} | 537.8 MB | 8.0 MB | 0.001 | 0.001 | 0.001 | 0.001 |
| | | | merged | [7.9 K (100.0 %)] | **537.8 MB** | 8.0 MB | 0.002 | 0.001 | 0.001 | 0.001 |
| | | ColA (MLP) | unmerged | {136.1K (1733.4 %)} | 537.8 MB | 10.0 MB | 0.001 | 0.002 | 0.001 | 0.002 |
| | 8 | FT | | 7.9 K (100.0 %) | 545.8 MB | | 0.001 | | | |
| | | LoRA | | 6.4k (80.9 %) | 545.8 MB | | 0.002 | | | |
| | | ColA (Low Rank) | unmerged | {6.4k (80.9 %)} | 537.8 MB | 8.0 MB | 0.001 | 0.001 | 0.001 | 0.001 |
| | | | merged | [6.4k (80.9 %)] | 537.8 MB | 8.0 MB | 0.002 | 0.001 | 0.001 | 0.001 |
| | | ColA (Linear) | unmerged | {7.9 K (100.0 %)} | 537.8 MB | 8.0 MB | 0.001 | 0.001 | 0.001 | 0.001 |
| | | | merged | [7.9 K (100.0 %)] | **537.8 MB** | 8.0 MB | 0.001 | 0.001 | 0.001 | 0.001 |
| | | ColA (MLP) | unmerged | {136.1K (1733.4 %)} | 537.8 MB | 10.0 MB | 0.001 | 0.002 | 0.001 | 0.002 |
| | 32 | FT | | 7.9 K (100.0 %) | 545.8 MB | | 0.001 | | | |
| | | LoRA | | 6.4k (80.9 %) | 545.8 MB | | 0.001 | | | |
| | | ColA (Low Rank) | unmerged | {6.4k (80.9 %)} | 537.8 MB | 8.0 MB | 0.002 | 0.013 | 0.001 | 0.001 |
| | | | merged | [6.4k (80.9 %)] | 537.8 MB | 8.0 MB | 0.001 | 0.001 | 0.002 | 0.002 |
| | | ColA (Linear) | unmerged | {7.9 K (100.0 %)} | 537.8 MB | 8.0 MB | 0.001 | 0.001 | 0.001 | 0.001 |
| | | | merged | [7.9 K (100.0 %)] | **537.8 MB** | 8.0 MB | 0.001 | 0.001 | 0.002 | 0.001 |
| | | ColA (MLP) | unmerged | {136.1K (1733.4 %)} | 537.8 MB | 10.0 MB | 0.001 | 0.002 | 0.001 | 0.002 |
| MLP, $M=3$ | 1 | FT | | 136.1 K (100.0 %) | 547.8 MB | | 0.001 | | | |
| | | LoRA | | 12.5 K (9.2 %) | 545.8 MB | | 0.001 | | | |
| | | ColA (Low Rank) | unmerged | {12.5 K (9.2 %)} | 545.8 MB | 0B | 0.001 | 0.001 | 0.002 | 0.001 |
| | | | merged | [12.5 K (9.2 %)] | 545.8 MB | 0B | 0.003 | 0.001 | 0.003 | 0.002 |
| | | ColA (Linear) | unmerged | {136.1 K (100.0 %)} | 547.8 MB | 0B | 0.002 | 0.001 | 0.003 | 0.001 |
| | | | merged | [136.1 K (100.0 %)] | **545.8 MB** | 2.0 MB | 0.002 | 0.001 | 0.002 | 0.001 |
| | | ColA (MLP) | unmerged | {350.2 K (257.4 %)} | 547.8 MB | 2.0 MB | 0.002 | 0.002 | 0.003 | 0.002 |
| | 8 | FT | | 136.1 K (100.0 %) | 547.8 MB | | 0.001 | | | |
| | | LoRA | | 12.5 K (9.2 %) | 545.8 MB | | 0.002 | | | |
| | | ColA (Low Rank) | unmerged | {12.5 K (9.2 %)} | 545.8 MB | 0B | 0.002 | 0.001 | 0.002 | 0.001 |
| | | | merged | [12.5 K (9.2 %)] | 545.8 MB | 0B | 0.002 | 0.001 | 0.003 | 0.001 |
| | | ColA (Linear) | unmerged | {136.1 K (100.0 %)} | 547.8 MB | 0B | 0.002 | 0.001 | 0.002 | 0.002 |
| | | | merged | [136.1 K (100.0 %)] | **545.8 MB** | 2.0 MB | 0.002 | 0.001 | 0.003 | 0.002 |
| | | ColA (MLP) | unmerged | {350.2 K (257.4 %)} | 547.8 MB | 2.0 MB | 0.002 | 0.002 | 0.003 | 0.003 |
| | 32 | FT | | 136.1 K (100.0 %) | 547.8 MB | | 0.001 | | | |
| | | LoRA | | 12.5 K (9.2 %) | 545.8 MB | | 0.002 | | | |
| | | ColA (Low Rank) | unmerged | {12.5 K (9.2 %)} | 545.8 MB | 0B | 0.002 | 0.001 | 0.002 | 0.001 |
| | | | merged | [12.5 K (9.2 %)] | 545.8 MB | 0B | 0.002 | 0.001 | 0.003 | 0.002 |
| | | ColA (Linear) | unmerged | {136.1 K (100.0 %)} | 547.8 MB | 0B | 0.002 | 0.001 | 0.002 | 0.001 |
| | | | merged | [136.1 K (100.0 %)] | **545.8 MB** | 2.0 MB | 0.002 | 0.001 | 0.003 | 0.001 |
| | | ColA (MLP) | unmerged | {350.2 K (257.4 %)} | 547.8 MB | 2.0 MB | 0.003 | 0.002 | 0.004 | 0.003 |
| CNN, $M=5$ | 1 | FT | | 155.5 K (100.0 %) | 593.8 MB | | 0.003 | | | |
| | | LoRA | | 44.2 K (2.8 %) | 721.8 MB | | 0.006 | | | |
| | | ColA (Low Rank) | unmerged | {44.2 K (2.8 %)} | 719.8 MB | 2.0 MB | 0.009 | 0.002 | 0.005 | 0.002 |
| | | | merged | [44.2 K (2.8 %)] | 591.8 MB | 132.0 MB | 0.011 | 0.002 | 0.005 | 0.004 |
| | | ColA (Linear) | unmerged | {155.5 K (100.0 %)} | 591.8 MB | 22.0 MB | 0.007 | 0.003 | 0.005 | 0.002 |
| | | | merged | [155.5 K (100.0 %)] | **591.8 MB** | 22.0 MB | 0.009 | 0.002 | 0.005 | 0.002 |
| | | ColA (MLP) | unmerged | {538.1 K (34.6 %)} | 593.8 MB | 4.0 MB | 0.007 | 0.003 | 0.006 | 0.002 |
| | 8 | FT | | 155.5 K (100.0 %) | 615.8 MB | | 0.004 | | | |
| | | LoRA | | 44.2 K (2.8 %) | 859.8 MB | | 0.004 | | | |
| | | ColA (Low Rank) | unmerged | {44.2 K (2.8 %)} | 725.8 MB | 6.0 MB | 0.008 | 0.026 | 0.005 | 0.002 |
| | | | merged | [44.2 K (2.8 %)] | 593.8 MB | 138.0 MB | 0.007 | 0.002 | 0.006 | 0.005 |
| | | ColA (Linear) | unmerged | {155.5 K (100.0 %)} | 613.8 MB | 2.0 MB | 0.013 | 0.004 | 0.005 | 0.002 |
| | | | merged | [155.5 K (100.0 %)] | **593.8 MB** | 22.0 MB | 0.021 | 0.003 | 0.006 | 0.002 |
| | | ColA (MLP) | unmerged | {538.1 K (34.6 %)} | 617.8 MB | 2.0 MB | 0.016 | 0.007 | 0.007 | 0.003 |
| | 32 | FT | | 155.5 K (100.0 %) | 615.8 MB | | 0.003 | | | |
| | | LoRA | | 44.2 K (2.8 %) | 881.8 MB | | 0.005 | | | |
| | | ColA (Low Rank) | unmerged | {44.2 K (2.8 %)} | 765.8 MB | 4.0 MB | 0.035 | 0.022 | 0.005 | 0.002 |
| | | | merged | [44.2 K (2.8 %)] | 633.8 MB | 266.0 MB | 0.022 | 0.006 | 0.005 | 0.004 |
| | | ColA (Linear) | unmerged | {155.5 K (100.0 %)} | 633.8 MB | 22.0 MB | 0.019 | 0.005 | 0.006 | 0.002 |
| | | | merged | [155.5 K (100.0 %)] | **633.8 MB** | 2.0 MB | 0.020 | 0.005 | 0.006 | 0.002 |
| | | ColA (MLP) | unmerged | {538.1 K (34.6 %)} | 695.8 MB | 80.0 MB | 0.012 | 0.017 | 0.007 | 0.002 |

Table 16: Computation evaluation of CLM task with user collaboration, GPT-2 model, and Dolly dataset, where the number of auxiliary models $M = 12$ and the number of users $K = 8$.

| Batch Size | Method | | Trainable Parameters | Memory | | Run Time (s) Offload to CPU | | Offload to GPU | |
|---|---|---|---|---|---|---|---|---|---|
| | | | | Base | Offload | Base | Offload | Base | Offload |
| 1 | Low Rank | unmerged | 8 × {294.9 K} | 1.3 GB | 10.0 MB | 0.073 | 0.004 | 0.040 | 0.004 |
| | | merged | 8 × [294.9 K] | 1.3 GB | 70.0 MB | 0.075 | 0.003 | 0.050 | 0.007 |
| | Low Rank-Linear | unmerged | 4 × {294.9 K}, 4 × {21.2 M} | 1.6 GB | 722.0 MB | 0.061 | 0.017 | 0.035 | 0.002 |
| | | merged | 4 × [294.9 K], 4 × [21.2 M] | **1.3 GB** | 1.1 GB | 0.064 | 0.004 | 0.053 | 0.006 |
| | Low Rank-MLP | unmerged | 4 × {294.9 K}, 4 × {8.7 M} | 1.4 GB | 196.0 MB | 0.066 | 0.012 | 0.041 | 0.005 |
| 8 | Low Rank | unmerged | 8 × {294.9 K} | 3.1 GB | 18.0 MB | 0.190 | 0.013 | 0.075 | 0.022 |
| | | merged | 8 × [294.9 K] | 3.1 GB | 24.0 MB | 0.095 | 0.010 | 0.051 | 0.019 |
| | Low Rank-Linear | unmerged | 4 × {294.9 K}, 4 × {21.2 M} | 3.6 GB | 600.0 MB | 0.135 | 0.031 | 0.065 | 0.019 |
| | | merged | 4 × [294.9 K], 4 × [21.2 M] | **3.1 GB** | 1.1 GB | 0.129 | 0.020 | 0.050 | 0.023 |
| | Low Rank-MLP | unmerged | 4 × {294.9 K}, 4 × {8.7 M} | 3.2 GB | 254.0 MB | 0.178 | 0.024 | 0.080 | 0.024 |
| 32 | Low Rank | unmerged | 8 × {294.9 K} | 9.0 GB | 18.0 MB | 1.000 | 0.017 | 0.205 | 0.032 |
| | | merged | 8 × [294.9 K] | 9.0 GB | 40.0 MB | 0.899 | 0.017 | 0.079 | 0.042 |
| | Low Rank-Linear | unmerged | 4 × {294.9 K}, 4 × {21.2 M} | 9.4 GB | 790.0 MB | 1.059 | 0.034 | 0.186 | 0.038 |
| | | merged | 4 × [294.9 K], 4 × [21.2 M] | **9.0 GB** | 1.1 GB | 0.971 | 0.030 | 0.075 | 0.017 |
| | Low Rank-MLP | unmerged | 4 × {294.9 K}, 4 × {8.7 M} | 9.2 GB | 58.0 MB | 1.076 | 0.031 | 0.212 | 0.037 |

Table 17: Computation evaluation of CLM task with user collaboration, Llama-2 $(Q, V)$, and Dolly dataset, where the number of auxiliary models $M = 64$ and the number of users $K = 8$. The auxiliary models of Llama-2 $(Q, V)$ include query and value projection layers.

| Batch Size | Method | | Trainable Parameters | Memory | | Run Time (s) Offload to CPU | | Offload to GPU | |
|---|---|---|---|---|---|---|---|---|---|
| | | | | Base | Offload | Base | Offload | Base | Offload |
| 1 | Low Rank | unmerged | 8 × {4.2 M} | 14.1 GB | 128.0 MB | 0.244 | 0.004 | 0.202 | 0.005 |
| | | merged | 8 × [4.2 M] | 14.0 GB | 1.1 GB | 1.655 | 0.003 | — | |
| | Low Rank-Linear | unmerged | 4 × {4.2 M}, 4 × {1.1 B} | 30.1 GB | > 7.9 GB | 0.252 | 0.223 | — | |
| | | merged | 4 × [4.2 M], 4 × [1.1 B] | **14.0 GB** | > 34.0 GB | 1.737 | 0.038 | — | |
| | Low Rank-MLP | unmerged | 4 × {4.2 M}, 4 × {103.0 M} | 15.6 GB | 1.7 GB | 0.216 | 0.011 | 0.356 | 0.005 |
| 8 | Low Rank | unmerged | 8 × {4.2 M} | 20.7 GB | 256.0 MB | 0.735 | 0.013 | 0.383 | 0.021 |
| | | merged | 8 × [4.2 M] | 20.8 GB | 828.0 MB | 1.283 | 0.013 | — | |
| | Low Rank-Linear | unmerged | 4 × {4.2 M}, 4 × {1.1 B} | 36.8 GB | > 11.2 GB | 0.724 | 0.301 | — | |
| | | merged | 4 × [4.2 M], 4 × [1.1 B] | **20.8 GB** | > 27.2 GB | 1.439 | 0.111 | — | |
| | Low Rank-MLP | unmerged | 4 × {4.2 M}, 4 × {103.0 M} | 22.6 GB | 3.1 GB | 0.844 | 0.030 | 0.465 | 0.018 |
| 32 | Low Rank | unmerged | 8 × {4.2 M} | 42.8 GB | 256.0 MB | 9.243 | 0.035 | — | |
| | | merged | 8 × [4.2 M] | 42.7 GB | 406.0 MB | 6.316 | 0.030 | — | |
| | Low Rank-Linear | unmerged | 4 × {4.2 M}, 4 × {1.1 B} | | | | | — | |
| | | merged | 4 × [4.2 M], 4 × [1.1 B] | **42.7 GB** | > 5.3 GB | 6.569 | 0.231 | — | |
| | Low Rank-MLP | unmerged | 4 × {4.2 M}, 4 × {103.0 M} | 44.6 GB | > 3.4 GB | 9.216 | 0.065 | — | |

Table 18: Computation evaluation of CLM task with user collaboration, Llama-2 (All), and Dolly dataset, where the number of auxiliary models $M = 224$ and the number of users $K = 8$. The auxiliary models of Llama-2 (All) include all projection layers.

| Batch Size | Method | | Trainable Parameters | Memory | | Run Time (s) Offload to CPU | | Offload to GPU | |
|---|---|---|---|---|---|---|---|---|---|
| | | | | Base | Offload | Base | Offload | Base | Offload |
| 1 | Low Rank | unmerged | 8 × {20.0 M} | 14.6 GB | 962.0 MB | 0.627 | 0.004 | 0.305 | 0.002 |
| | | merged | 8 × [20.0 M] | 14.1 GB | 4.5 GB | 8.918 | 0.003 | — | |
| | Low Rank-Linear | unmerged | 4 × {20.0 M}, 4 × {6.5 B} | | | | | — | |
| | | merged | 4 × [20.0 M], 4 × [6.5 B] | **14.1 GB** | > 33.9 GB | 8.959 | 0.051 | 0.356 | 0.003 |
| | Low Rank-MLP | unmerged | 4 × {20.0 M}, 4 × {502.7 M} | 22.2 GB | 11.7 GB | 0.635 | 0.017 | — | |
| 8 | Low Rank | unmerged | 8 × {20.0 M} | 21.6 GB | 1.2 GB | 4.583 | 0.015 | 1.133 | 0.007 |
| | | merged | 8 × [20.0 M] | 20.9 GB | > 27.1 GB | 7.956 | 0.016 | — | |
| | Low Rank-Linear | unmerged | 4 × {20.0 M}, 4 × {6.5 B} | | | | | — | |
| | | merged | 4 × [20.0 M], 4 × [6.5 B] | **20.9 GB** | 4.0 GB | 6.949 | 0.171 | — | |
| | Low Rank-MLP | unmerged | 4 × {20.0 M}, 4 × {502.7 M} | 29.7 GB | 15.9 GB | 5.588 | 0.044 | 1.311 | 0.008 |
| 32 | Low Rank | unmerged | 8 × {20.0 M} | 44.2 GB | 1.2 GB | 41.893 | 0.058 | — | |
| | | merged | 8 × [20.0 M] | 42.8 GB | 3.5 GB | 30.033 | 0.060 | — | |
| | Low Rank-Linear | unmerged | 4 × {20.0 M}, 4 × {6.5 B} | | | | | — | |
| | | merged | 4 × [20.0 M], 4 × [6.5 B] | **42.8 GB** | > 5.2 GB | 30.818 | 0.440 | — | |
| | Low Rank-MLP | unmerged | 4 × {20.0 M}, 4 × {502.7 M} | | | | | — | |

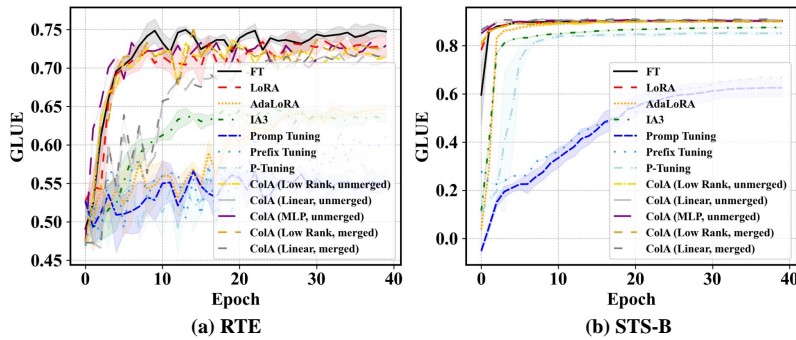

**(a) RTE**  **(b) STS-B**

Figure 14: Learning curves of (a) RTE and (b) STS-B datasets of SC task and GLUE metric

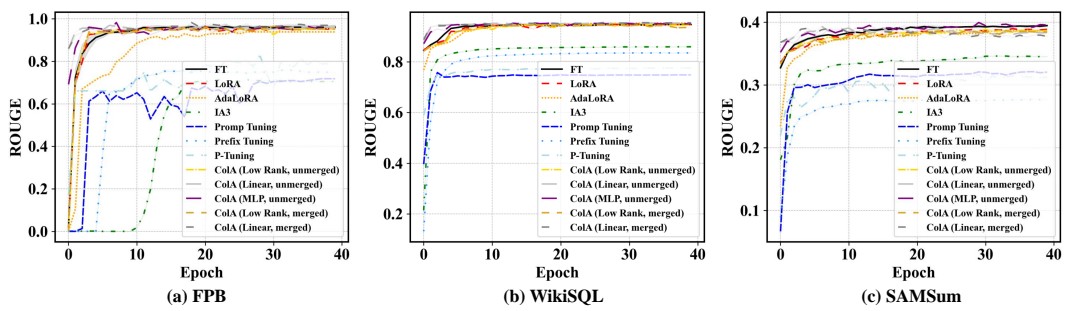

**(a) FPB**  **(b) WikiSQL**  **(c) SAMSum**

Figure 15: Learning curves of (a) FPB (b) WikiSQL, and (c) SAMSum datasets of S2S task and ROUGE (Longest) metric.

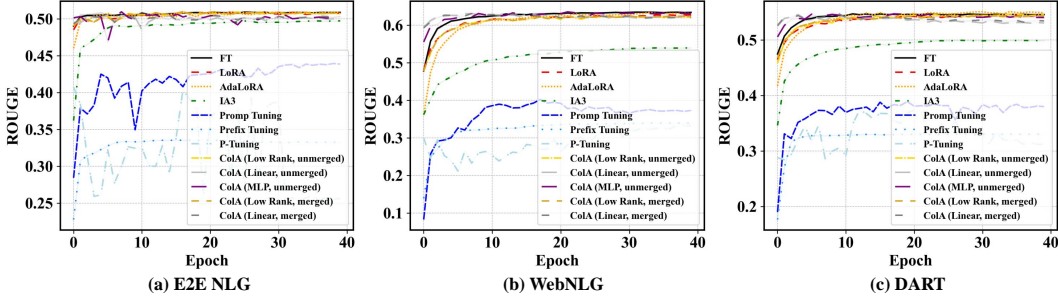

**(a) E2E NLG**  **(b) WebNLG**  **(c) DART**

Figure 16: Learning curves of (a) E2E NLG (b) WebNLG, and (c) DART datasets of S2S task and ROUGE (Longest) metric.

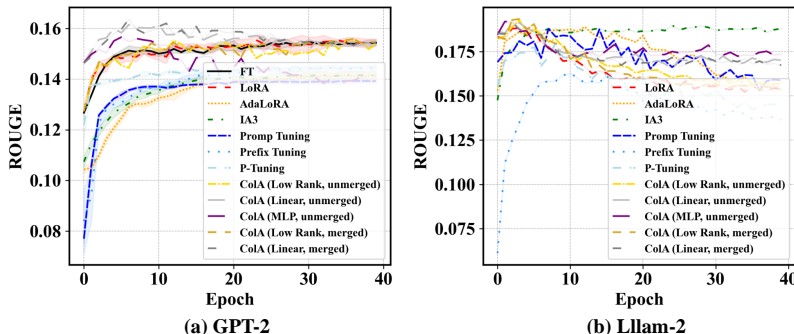

**(a) GPT-2**  **(b) Lllam-2**

Figure 17: Learning curves of (a) GPT-2 and (b) Llama-2 ($Q, V$) on Dolly dataset of CLM task and ROUGE (Longest) metric.

