# OpenReview forum: "ColA: Collaborative Adaptation with Gradient Learning"
_ICLR.cc/2024/Conference — Submitted to ICLR 2024_

### Official Review · Reviewer_cUyw · 2023-10-30

**Soundness:** 3 good
**Presentation:** 2 fair
**Contribution:** 2 fair
**Rating:** 3
**Confidence:** 3

**Summary:**

The paper proposes a fine-tuning framework to improve the storage efficiency of parameter-efficient fine-tuning methods. Specifically, the proposed method proposes Gradient Learning to decouple the computation of the gradient of the model weights and the gradient of hidden features. For example, the main forward and backward pass through the model only calculates the gradient of the hidden features, and the calculation of the gradient of the model weights can be off-loaded to another low-cost device with an auxiliary quadratic loss. The main benefit is that the main forward and backward pass no longer needs to calculate the gradient of the model weights, which supposedly saves storage on the main device.

**Strengths:**

* **Gradient Learning is Novel**: the idea of decoupling the calculation of the gradient of the model weights and the gradient of hidden features seems novel. The paper also theoretically demonstrates the equivalence between the proposed decoupled update and the conventional update rules.

**Weaknesses:**

* **Why is the method parameter-free?**: even though the proposed model offloads the update of adapter weights to a different device, it does not make it parameter-free. It is not very convincing to claim the fine-tuning method to be parameter-free.

* **Actual memory footprint not clear**: while the model claims to save storage on the main device, e.g., the GPU, the paper does not report the actual memory footprint during the forward and backward passes on the GPU. Compared to the small number of trainable parameters in LORA/adapters, the hidden feature maps and their gradients utilize the most memory. Offloading the gradient computation of the trainable parameters seems to only marginally improve memory usage on the main device. It would be great to see how much memory is actually saved on the main device by using the proposed decoupled update.

* **Time efficiency not reported**: the method involves an update on a second low-cost device, e.g., a CPU. The paper does not discuss the impact on training time efficiency.

* **Some method components are not very relevant**: the discussion on parameter merging is not very relevant to the main proposed method. It is also not clear which experiments underscore the benefits of user collaboration as claimed in the contribution. Overall, the components of parameter merging and user collaboration seem tangential to the proposed method and are not well analyzed through experiments.


Minor:
* The number of trainable parameters of Co1A in Table 2 is much smaller than that of LORA. Is this a typo?

**Questions:**

* Could the authors comment on the parameter-free property of the method?

* Could the authors report on the actual memory usage and savings on the main device?

* Could the authors provide a discussion, preferably quantitatively, of the impact on the time efficiency?

---

> ### Author Response · Authors · 2023-11-18
> **Response**
>
> Thank you for your constructive feedback. In response to your comments, we have refined our method, revised the paper, and addressed your concerns as outlined below:
> 1. Why is the method parameter-free?: even though the proposed model offloads the update of adapter weights to a different device, it does not make it parameter-free. It is not very convincing to claim the fine-tuning method to be parameter-free. Could the authors comment on the parameter-free property of the method ?
> > The term 'parameter-free' in our context underscores the absence of auxiliary parameters during the back-propagation of the base model. Initially, utilizing the 'detach' technique would exclude any auxiliary models from the back-propagation process. However, to precisely align with the computations of LoRA and classical back-propagation, we realized that it is necessary to utilize the 'parameter merging' mechanism during training as demonstrated in Algorithm 1. For a detailed explanation, please refer to the main response.
> 2. Could the authors report on the actual memory usage and savings on the main device? Could the authors provide a discussion, preferably quantitatively, of the impact on the time efficiency? Actual memory footprint not clear ... The paper does not discuss the impact on training time efficiency.
> > We conducted a comprehensive quantitative evaluation of computational costs for all feasible experiments. The results, outlined from Table 9 to 17 of the Appendix, demonstrate a significant reduction in computation space bottleneck during the back-propagation of the base model through our method. Moreover, our approach allows for a reduction in run time when offloading to an additional GPU. Further discussion on the computation can be found in the main response and specific comparisons in terms of memory usage and run time relative to baselines can be found in the revised paper.
> 3. Some method components are not very relevant: the discussion on parameter merging is not very relevant to the main proposed method. It is also not clear which experiments underscore the benefits of user collaboration as claimed in the contribution. Overall, the components of parameter merging and user collaboration seem tangential to the proposed method and are not well analyzed through experiments.
> >  Parameter merging used in our refined method plays a crucial role in reducing the computation space bottleneck during training and also impacts memory consumption during inference, as evidenced in Algorithm 1 and Table 1. User collaboration is also essential to our method's applicability in Fine-Tuning-as-a-Service. As demonstrated in Table 15 to 17, ColA (merged) consumes the same amount of memory for a given batch size, regardless of the size of auxiliary models and the number of users, because the computation of all auxiliary models has been offloaded to a separate device.
> To address your concerns regarding user collaboration, we have now incorporated it into Algorithm 1. To facilitate Fine-Tuning-as-a-Service in a computationally scalable manner, it is impractical to train $K$ distinct large models for $K$ users. Our ColA (unmerged) method allows hosting a single large model to provide fine-tuning services for $K$ users, each training separate adapters. Moreover, ColA (merged) can leverage local data and computation resources for collaborative fine-tuning of the large model.
> 4. The number of trainable parameters of Co1A in Table 2 is much smaller than that of LORA. Is this a typo?
> > In Table 2, the process of fine-tuning a classification model involves creating a randomly initialized output head. The peft package from Huggingface originally trains these newly introduced parameters directly without prior fine-tuning, and also double count them.  In our initial implementation of ColA (Low Rank), the number of parameters used is lower due to the application of low-rank approximation on this output head. We have addressed the issue of double counting in the peft package and used a linear layer for this specific output head for all ColA methods. This adjustment ensures that the number of trainable parameters of ColA (Low Rank) and LoRA are now precisely aligned. The updated results reflecting these changes are presented in the revised paper.

---

### Official Review · Reviewer_UBHP · 2023-10-31

**Soundness:** 3 good
**Presentation:** 2 fair
**Contribution:** 2 fair
**Rating:** 5
**Confidence:** 2

**Summary:**

This paper proposes a new fine-tuning method to reduce GPU computational and memory costs. Specifically, the method offloads the gradient update to the auxiliary variable from a GPU to a CPU. The authors provide theoretical analysis to justify its correctness. Based on this new learning method, the authors further introduce a collaborative learning framework. Experiments on RoBERTa and BART demonstrate the effectiveness of the proposed method.

**Strengths:**

-	The proposed method is simple and easy to implement.
-	Efficient fine-tuning is an important topic.

**Weaknesses:**

- The motivation and advantages of moving gradient update to CPU are unclear.
- The relationship between the proposed gradient learning and collaborative adaption is unclear.
- The experiments do not include collaborative adaption.

**Questions:**

-	As far as I understand, the key contribution of the proposed method is to offload the update for the auxiliary variables w to a CPU, rather than a GPU. After reading the paper, it is still unclear to me why this is desirable. Can the authors explain the motivation and advantages of doing so?
-	In Proposition 2, the model is assumed to be linear. How is this result related to the case considered in the paper where the model is a pretrained neural network?
-	It is unclear how the proposed gradient learning method related to collaborative adaptation. Why is gradient learning important for collaborative adaptation? What is the problem setup of collaborative adaptation in this paper?
-	The paper focuses on collaborative adaptation, but the experiments do not seem to include collaborative adaptation.
-	The communication cost between GPU and CPU is unclear.

---

> ### Author Response · Authors · 2023-11-18
> **Response**
>
> Thank you for your constructive feedback. In response to your comments, we have refined our method, revised the paper, and addressed your concerns as outlined below:
> 1. The motivation and advantages of moving gradient update to CPU are unclear. As far as I understand, the key contribution of the proposed method is to offload the update for the auxiliary variables w to a CPU, rather than a GPU. After reading the paper, it is still unclear to me why this is desirable. Can the authors explain the motivation and advantages of doing so?
> > The primary challenge in leveraging GPU for deep learning acceleration lies in its limited yet valuable memory capacity. Larger memory enables the training of larger models with a larger batch size. For instance, large model Llama-2 7B, even in 16-bit float precision, demands 12.6 GB of memory for mere GPU loading during inference. As illustrated in Tables 13, training such models requires substantially more memory, exceeding the capacity of a 48 GB GPU, even with a batch size of one. Large batch sizes, while beneficial for gradient estimation and model convergence, further exacerbate memory constraints. This challenge is evident even in advanced methods like LoRA, which struggle to fit in a 48 GB GPU with a batch size of 32. A notable approach to mitigate this issue is the Zero-Offload method utilized in DeepSpeed, which involves offloading computed GPU gradients to the CPU, so that the state information of adaptive optimizers like Adam can be saved on CPU, thereby conserving GPU memory (as described in https://www.microsoft.com/en-us/research/blog/deepspeed-extreme-scale-model-training-for-everyone/). To quote from the section "Learn how ZeRO-Offload enables multi-billion parameter training on a single GPU": `Gradients, on the other hand, are computed and averaged using reduce-scatter on the GPUs during the backward pass, and each data-parallel process then offloads the averaged gradients belonging to its partition to the CPU memory.' Our method aims to address the memory challenge by offloading the entire computation of gradient of parameters to the CPU, and also naturally allowing optimizer state information to reside on CPU. Table 1 demonstrates how our method, ColA (merge), significantly reduces memory usage by transferring gradient computations to an additional device. Furthermore, it can also increase the effective batch size by adjusting the adaptation interval. Further discussion on the computation can be found in the main response and specific comparisons in terms of memory usage and run time relative to baselines can be found in the revised paper.
> 2. In Proposition 2, the model is assumed to be linear. How is this result related to the case considered in the paper where the model is a pretrained neural network?
> > We clarify that Proposition 2 is applicable to the intermediate layers of the network. The term 'fine-tuned layers' in this context specifically refers to certain intermediate layers, not the entire neural network.
> 3. The relationship between the proposed gradient learning and collaborative adaption is unclear. It is unclear how the proposed gradient learning method related to collaborative adaptation. Why is gradient learning important for collaborative adaptation? What is the problem setup of collaborative adaptation in this paper?
> > To address your concerns, we have now incorporated user collaboration into Algorithm 1. To facilitate Fine-Tuning-as-a-Service in a computationally scalable manner, it is impractical to train $K$ distinct large models for $K$ users. Our ColA (unmerged) method allows hosting a single large model to provide fine-tuning services for $K$ users, each training separate adapters. Moreover, ColA (merged) can leverage local data and computation resources for collaborative fine-tuning of the large model.
> 4. The experiments do not include collaborative adaption. The paper focuses on collaborative adaptation, but the experiments do not seem to include collaborative adaptation.
> > Initially, our experiments on user collaboration were detailed in the Appendix. To address your concerns, we have now moved the related section in Appendix to the main body of the paper.
> 5. The communication cost between GPU and CPU is unclear.
> >  We demonstrate the run time of the proposed method from Table 9 to 17 of the Appendix. The results demonstrate that the run time scales with batch size when we offload the computation to a CPU. However, offloading computation to an additional GPU significantly reduces run time. For a comprehensive discussion on computation costs, please refer to the main response and the revised paper.

---

### Official Review · Reviewer_NXJy · 2023-11-01

**Soundness:** 3 good
**Presentation:** 3 good
**Contribution:** 3 good
**Rating:** 6
**Confidence:** 4

**Summary:**

This work proposed a method termed ColA, for efficiently adapting a pretrained model for a downstream task. In particular, the proposed method assumes an auxiliary set of parameters, which are used to parametrize auxiliary functions, that take the hidden representation of a layer and transform it by adding a computed delta shift, before it is fed as input to the next layer.

To avoid having to store and update the parameters of these light-weight auxiliary models on the GPU, this work proposes a "Gradient Offloading" strategy, wherein gradients with respect to the change in hidden representations are offloaded to the CPU, and a gradient update with respect to the auxiliary parameters model parameters is computed (and potentially updated) offline on the CPU. The proposed ColA method is model-agnostic, and can be adapted to any set of auxiliary models. Moreover, the weights of the pretrained model themselves are never altered during fine-tuning.

The authors propose ColA for offering Fine-Tuning as a Service (FTaaS) in commercial settings. The idea is that ColA can be used to provide mass personalization of foundation model fine-tuning for users. Users can own their own set of auxiliary parameters used to update the hidden representations of the pretrained network during the forward pass, and can share in the computational update of these auxiliary parameters in a federated-learning style manner.

Extensive numerical experiments are provided on the following tasks
* Sequence Classification -> RoBERTa (base) on GLUE
* Sequence to Sequence Modelling -> BART (base) on Financial Phrase Bank, WikiSQL, DART
* Causal Language Modelling -> GPT2 with instruction tuning on Dolly

comparing the following methods
* full fine-tuning
* LoRA
* AdaLoRA
* IA3
* Prompt Tuning
* Prefix Tuning
* P-Tuning

and comparing the following set of auxiliary weights for their proposed ColA method
* Low Rank
* Linear
* MLP

**Strengths:**

### Originality
* Novelty of ColA: The main innovation in ColA is the ability to compute updates to auxiliary parameters offline. Previous approaches for efficient model adaptation include: Fine-tuning Adapter layers, which place learnable layers in-between existing learnable layers; Low-Rank Adaptation (LoRA), which introduce two low-rank matrices to parametrize the updates of pretrained weigh matrices; and Prefix Tuning, which prepends sequence of learnable tokens as input to the network. Conceptually, ColA can be used with many of these strategies, and unlocks the ability to compute the auxiliary parameter updates offline.

### Clarity
* The paper is well written and sufficiently easy to follow given all the technical components introduced.

### Significance
* The proposed Fine-Tuning as a Service (FTaaS) framework is interesting in my personal opinion, and likely to be of increasing relevance to the ICLR community. Of notable interest is the intersection of this framework with Federated Learning, which has also been proposed for model personalization, but perhaps in more general settings (i.e., without placing constraints on adapting a small set of auxiliary parameters for a large frozen foundation model).

**Weaknesses:**

Weaknesses
* Not clear how to align proposed method with other optimization strategies (i.e., beyond gradient descent)
* Still need to forward/backward propagate the model K times for K users; i.e., the decoupled gradient computation and adaptation does not address this issue
* To compute the change in hidden state at some layer $m < M$, you need to have first computed the hidden state at layer $m-1$. Since this computation is carried out on the GPU (server); it appears as though you either need to eventually send the local model to the server and place it on device, or you have numerous iteration rounds between the server and client to compute a single forward pass, and send the targets for an offline update of the auxiliary parameters.
* While it is claimed that the offline update is equivalent to the online update of auxiliary parameters, this does not seem to be the case in practice (based on training curves, gradient magnitudes, and performance compared to non-offloaded computations on considered tasks)
* It is not clear to me how memory is actually saved by offloading gradients with respect to the small set of auxiliary parameters to the CPU, and to what degree training is slowed down due to this offloading.


* Minor error; page 4. I believe you mean the gradient of the auxilary parameters $\nabla w_{1:M}$
* Minor error; page 5. I believe you mean $\nabla \delta h^t_m = g^t_m(x^t_m)$

**Questions:**

* How much memory do you actually save by offloading gradients with respect to the small set of auxiliary parameters, and what is the increase in training time due to the offloading and recompilation phase? Please include logs comparing ColA to baselines in terms of memory, forward time, backward time, and number of host-device transfers.
* Could you please clarify why the ColA LoRA updates (the best performing CoLA setting in Sequence Classification tasks) are not strictly equivalent to non-offloaded LoRA updates?

---

> ### Author Response · Authors · 2023-11-18
> **Response**
>
> Thank you for your constructive feedback. In response to your comments, we have refined our method, revised the paper, and addressed your concerns as outlined below:
> 1. Not clear how to align proposed method with other optimization strategies (i.e., beyond gradient descent)
> > Our work generalizes functional gradient descent in the context of deep learning. The auxiliary model is model-agnostic, serving as a functional to enhance the performance of the original network. This enhancement is achieved by leveraging the gradient or the so-called 'pseudo-residual' of the hidden representations.
> 2. Still need to forward/backward propagate the model K times for K users; i.e., the decoupled gradient computation and adaptation does not address this issue.
> > Contrary to the raised concern, our method requires only a single forward and backward pass for $K$ users. This efficiency is due to our approach of back propagating the gradient of hidden representations. By concatenating data from K users into a single batch, each adapter can simultaneously route corresponding samples to different adapters in parallel. We implement a `Router' class, which effectively organizes this process within a virtually parallel loop, but still we only require one forward and backward pass for a single batch of data aggregated from $K$ users.
> 3. To compute the change in hidden state at some layer $m < M$, you need to have first computed the hidden state at layer $m-1$. Since this computation is carried out on the GPU (server); it appears as though you ...
> > We do not need numerous iteration rounds for a single forward pass. In both ColA (unmerged) and ColA (merged) methods, it is only necessary for users to upload their fine-tuned auxiliary models to the server for each forward and backward pass. After the server back propagates on the base model, the hidden representations and their gradients are transferred back to the users for local updates of the auxiliary models.
> 4. While it is claimed that the offline update is equivalent to the online update of auxiliary parameters, this does not seem to be the case in practice (based on training curves, gradient magnitudes, and performance compared to non-offloaded computations on considered tasks)
> > We have resolved the previously identified discrepancy between offline and online updates of auxiliary parameters. The root cause was the `detached' mechanism previously utilized, which ignored the gradient information of auxiliary models in intermediate layers. Our revised methods now accurately align with the gradient of LoRA and classical back-propagation. Further details are available in the main response.
> 5. It is not clear to me how memory is actually saved by offloading gradients with respect to the small set of auxiliary parameters to the CPU, and to what degree training is slowed down due to this offloading. How much memory ...
> > We conducted a comprehensive quantitative evaluation of computational costs for all feasible experiments. The results, outlined from Table 9 to 17 of the Appendix, demonstrate a significant reduction in computation space bottleneck during the back-propagation of the base model through our method. Moreover, our approach allows for a reduction in run time when offloading to an additional GPU. Further discussion on the computation can be found in the main response and specific comparisons in terms of memory usage and run time relative to baselines can be found in the revised paper.
> 6. Minor error; page 4. I believe you mean the gradient of the auxilary parameters. Minor error; page 5 ...
> > The errors on pages 4 and 5 have been corrected in the revised version.

---

### Author Response · Authors · 2023-11-18
**Main Response**

Thank you for your time and constructive comments. We have addressed all the comments below. The following major changes are included and highlighted in the revised paper. We hope the responses and revisions will be viewed favorably.
1. **Method** We have refined the implementation of our method to align accurately with LoRA and classical back-propagation in the computation of gradient of parameters. Our initial implementation, termed ColA (detached), utilized a 'detach' mechanism, inadvertently neglecting gradient information of auxiliary models in intermediate layers. To rectify this, we eliminated the 'detach' mechanism, maintaining the auxiliary model throughout back-propagation. However, this adjustment named as ColA (unmerged) reduces the computational benefit to solely offloading the computation of gradient of auxiliary parameters. Then, we discovered that by utilizing the `parameter merging' technique during training as demonstrated in Algorithm 1, our method named as ColA (merged) can further reduce the computation cost by offloading the entire auxiliary models to another device. This approach is also utilized in user collaboration to further decrease the computational cost. We have rerun all the experiments and present the new results in the revised paper.
2. **Experiments** To substantiate the aforementioned advancements, we have implemented a script named 'verify.py', which is included in the supplementary materials. This script conceptually validates our assertions regarding Eq. 5, Proposition 1, and the different variations of ColA (detached, unmerged, and merged). Notably, it shows that ColA (detached) matches the gradient of classical back-propagation only in the final layer, while ColA (unmerged) and ColA (merged) accurately match the gradient across all layers. Additionally, we have updated Table 1 to conceptually compare the computational costs of with these methods. Notably, ColA (merged) can reduce the cost of full fine-tuning by offloading the computation of gradient of parameters to another device. To our best knowledge, this has not been achieved in any existing methods. It indicates that our method can achieve the performance of full parameter training from scratch while reducing the computation space bottleneck. To corroborate this, we trained the MNIST and CIFAR10 datasets on Linear, MLP, and CNN models from scratch, as detailed in Section C.3 of the Appendix. The results demonstrate that LoRA yields suboptimal results due to low-rank approximation while our method can achieve the results of full-fine tuning, because we can train the model without any approximation.
3. **Computation** We conducted a comprehensive quantitative evaluation of computational costs for all feasible experiments on real devices, as suggested by the reviewers. The results, outlined from Table 8 to 16 of the Appendix, demonstrate a significant reduction in computation space bottleneck during the back-propagation of the base model through our method, while also indicating that run time can be decreased by offloading to an additional GPU.
    1. Our method, ColA (Low Rank), which matches LoRA exactly, consistently uses less computation space than LoRA during back propagation of the base model.
    2. Despite the limited size of auxiliary models only providing a small advantage over LoRA (2-4 GB in Table 12 and 13), our ColA (Linear, merged) can train the network without low-rank approximation and significantly reduce the computation space bottleneck. For example, as shown in Table 12, ColA (Linear, merged, batch size=8) requires 20.8 GB, compared to LoRA's 21.8 GB. Similarly, Table 13 shows that ColA (Linear, merged, batch size=32) uses 42.8 GB, while LoRA exceeds 48 GB and does not fit on our device. Notably, our method also requires less computation space than direct training on the base model. For instance, full fine-tuning in Table 13 does not fit in 48 GB device even at a batch size of one.
    3. As demonstrated from Table 15 to 17, ColA (merged) consumes the same amount of memory for a given batch size, regardless of the size of auxiliary models and the number of users, because the computation of all auxiliary models has been offloaded to a separate device.
    4. The results demonstrate that the run time scales with batch size when we offload the computation to a CPU. However, offloading computation to an additional GPU significantly reduces run time. It is foreseeable as the communication of tensors among GPUs are much faster than it between GPU and CPU. A pivotal aspect of our method is that the computation of gradient of model parameters can be decoupled from the classical back-propagation and distributed across multiple smaller devices with lower memory capacity. This is particularly desirable considering the affordability and availability of consumer-grade GPUs with smaller memory capacities, such as the 3090, in contrast to professional-grade GPUs like the H100.

---

> ### Author Response · Authors · 2023-11-18
> **Main Response (cont.)**
>
> Our work distinguishes itself from existing works that primarily focus on designing a parameter-efficient auxiliary models. We instead target the computational challenges inherent in classical learning algorithms. We question the prevailing assumption that the gradient of parameters and hidden representations must be computed concurrently. We respectfully hope the reviewer to acknowledge our contribution not only in the domain of Parameter Efficient Fine-Tuning but also to methodologically recognize the significance of our approach within the broader scope of learning and optimization.

---

### Meta-Review · Area_Chair_nz2r · 2023-12-10

**Metareview:**

This paper proposes a different fine-tuning method to reduce GPU computational and memory costs, using a strategy for apparent decoupling of main from auxiliary parameter computations.

The paper in its current writing is prone to overselling some parts, such as 'parameter free', when the meaning is no additional hyperparameters, imprecise terminology such as 'computational space', and typos such as 'Adpater' training. I would suggest to have the Llama2 results in the main paper.
I'm worried the hyperparameter selection of the proposed models vs baselines is not clear and potentially not fair. Appendix C only shows one learning rate for PEFT and for ColA, but doesn't explain how it was chosen to potentially favor one of the two. As the 'parameter free' claim is already in the abstract this needs to be stated more precisely. The theoretical motivation with functional gradient descent is not properly explained, which again hurts the contributions. Unfortunately I also don't see a very convincing argument in the paper why existing PeFT methods can not achieve the same goal of offloading the auxiliary parameters and computations away from the GPU, with the same effect on compute as desired (as to precisely state what parts of the classic backward pass on the main parameters can be skipped or not, and when. State clear impact in terms of Flops and memory). For instance, DeepSpeed offloading computes the same gradients (so original backwards) but slower with less memory, so doesn't fit the story here.

Overall, some reviewers liked the idea, however missed several aspects and clarify of presentation unfortunately, even after the response. Also we felt several concise questions by the reviewers were not very clearly answered. It's not clear to me if that was a language issue, but it affects the precision of the claims, both in the paper and the discussion phase.
Consensus among the reviewers is that it remains narrowly below the bar even after the discussion phase.
We hope the detailed feedback helps to strengthen the paper for a future occasion.

**Justification For Why Not Higher Score:**

The paper in current form is weak on rigorous explanations for almost all its main claims

**Justification For Why Not Lower Score:**

N/A

---

### Decision · Program_Chairs · 2024-01-16

Reject